# A hierarchical model for external electrical control of an insect, accounting for inter-individual variation of muscle force properties

Dai Owaki[1]*, Volker Dürr[2,3], Josef Schmitz[2,3]

[1]Department of Robotics, Graduate School of Engineering, Tohoku University, Sendai, Japan; [2]Department of Biological Cybernetics, Faculty of Biology, Bielefeld University, Bielefeld, Germany; [3]Centre for Cognitive Interaction Technology, Bielefeld University, Bielefeld, Germany

**Abstract** Cyborg control of insect movement is promising for developing miniature, high-mobility, and efficient biohybrid robots. However, considering the inter-individual variation of the insect neuromuscular apparatus and its neural control is challenging. We propose a hierarchical model including inter-individual variation of muscle properties of three leg muscles involved in propulsion (retractor coxae), joint stiffness (pro- and retractor coxae), and stance-swing transition (protractor coxae and levator trochanteris) in the stick insect *Carausius morosus*. To estimate mechanical effects induced by external muscle stimulation, the model is based on the systematic evaluation of joint torques as functions of electrical stimulation parameters. A nearly linear relationship between the stimulus burst duration and generated torque was observed. This stimulus-torque characteristic holds for burst durations of up to 500ms, corresponding to the stance and swing phase durations of medium to fast walking stick insects. Hierarchical Bayesian modeling revealed that linearity of the stimulus-torque characteristic was invariant, with individually varying slopes. Individual prediction of joint torques provides significant benefits for precise cyborg control.

**\*For correspondence:** owaki@tohoku.ac.jp

**Competing interest:** The authors declare that no competing interests exist.

## Editor's evaluation

This valuable work presents new results to characterize the relationship between electrical excitation and torque generation in stick insect joints. The evidence supporting this work is a series of torque-voltage measurements across individuals. The strength of evidence is compelling in supporting the outcomes.

## Introduction

Hybrid insect–computer robots (*Krause et al., 2011*; *Li and Sato, 2018*) represent cutting-edge approaches to develop robots with locomotor performances comparable to those of insects. With the advancement and diversity in micro-flexible and micro-printable electronics (*Rogers et al., 2010*; *Rich et al., 2021*), micro-mechanical fabrication, and micro-actuator technologies (*Kim et al., 2020*), such biohybrid, that is cyborg robots have been engineered to manipulate their gait and flight through electrical stimulation of target muscles in various insects, includings beetles (*Sato, 2009*; *Sato and Maharbiz, 2010*; *Sato et al., 2015*; *Cao et al., 2016*; *Vo Doan et al., 2018*; *Nguyen et al., 2020*; *Kosaka et al., 2021*), moths (*Sane et al., 2007*; *Bozkurt et al., 2009*; *Hinterwirth et al., 2012*; *Ando and Kanzaki, 2017*), and cockroaches (*Sanchez et al., 2015*). The advantage of biohybrid (cyborg)

**eLife digest** Hybrid insect-computer robots – an exciting fusion of biology and technology – herald a future of small, highly mobile and efficient devices. However, these robots require a way to control the movements of the insects, a task made complex due to the differences between different insects' nervous and muscle systems.

To bridge this gap, Owaki, Dürr and Schmitz studied the relationship between electrical stimulation of three leg muscles in the legs of stick insects and the resultant torque. To do these experiments, the scientists kept the body of the stick insects fixed and electrically stimulated one out of three leg muscles to produce walking-like movements.

The results of these electrical stimulations allowed Owaki, Dürr and Schmitz to propose a model that predicts the torque created in the insect's joints when different patterns of electrical stimulation are applied to a leg muscle. The researchers identified a near-linear relationship between the duration of the electrical stimulus and the resultant torque. Moreover, the slope of this linear relationship can be estimated for individual animals with a few measurements only. This finding refines the precision of the motor control required to build individually tuned biohybrid robots.

Investigating the precise control of insect biohybrid robots, particularly using stick insects, can lead to advancements in biohybrid robotics and enrich our understanding of insect locomotion.

Owaki, Dürr and Schmitz's insights could lead to the creation of adaptable and highly mobile devices with many applications, but key challenges need to be addressed. First, model testing must be implemented in free-walking insects, and the electrical stimuli must be refined to mimic natural neuromuscular signals more closely.

robots is that they do not require individual 'design', 'fabrication', and 'assembly' processes for each component because they use the body tissues of living insects (*Cao et al., 2014*). Moreover, cyborg robots have low power consumption, that is, a few milliwatts (*Sato and Maharbiz, 2010*). Although studies on insect cyborgs have demonstrated simple manufacturing and promising energy efficiency, they are still in the initial phase of development from the perspective of evaluating both feasibility and reliability of their control.

Perhaps, the greatest challenge in cyborg control comes with the inter-individual variability of animals. Past neurophysiological studies related to animal neural activity have discussed the failure of averaging-based approaches, in which a model formulated using the average data for a group cannot explain the characteristics of any individual in the group (*Golowasch et al., 2002*; *Schulz et al., 2006*). For example, variable and non-periodic patterns in feeding behavior of Aplysia have been reported to be subject to strong inter-individual variation (*Horn et al., 2004*; *Brezina et al., 2005*; *Zhurov et al., 2005*). In insect motor physiology, the prediction error of muscle models which are based on sample averages is very high (*Blümel et al., 2012a*) and may be halved using individual-specific model (*Blümel et al., 2012b*). At the level of leg movements, variability has been investigated in lobsters (*Thuma et al., 2003*) and stick insects (*Hooper et al., 2006*). The variability of whole-body locomotion arises from step parameter variation of single legs (*Theunissen and Dürr, 2013*) but also from variation of coupling strength among legs (*Dürr, 2005*). One possible approach for accounting for inter-individual variability in cyborg control of single-leg movement is to construct a feedback control system (*Cao et al., 2014*). Although the kinematics-control of joint angles (*Cao et al., 2014*) has exhibited remarkable performance, its applicability to the control of dynamic gaits, such as that for walking, is still controversial. Furthermore, insects have abundant control variables, that is, degrees of freedom in their actuators and sensors. At present, the number of control variables of current insect cyborgs has to be reduced owing to system implementation difficulties.

A promising approach to overcome the 'pitfalls' associated with averaging across individuals is to understand the underlying principles that govern inter-individual variability in insect motor control. Especially, the output characteristics of muscle are key for controlling the dynamics of movement: muscles convert neural activity into movement and then generate behavior from interactions with the environment. In conjunction with current models of muscle activation (*Harischandra et al., 2019*) and contraction dynamics (*Blümel et al., 2012c*), we can exploit experimental data to tell parameters that are strongly influenced by inter-individual variation as opposed to others that are common

characteristics. To this end, we employed a hierarchical modeling framework based on the Bayesian statistical analysis (*Watanabe, 2018*; *Gelman et al., 2013*) that explicitly accounts for inter-individual variation in experimental data. In particular, we applied a set of hierarchical Bayesian models with different combinations of common and individually varying parameters and mathematically evaluated their prediction performance.

The main objective of this study was to systematically evaluate how muscle force and corresponding joint torques depend on external electrical stimulation, as a fundamental pre-requisite for precise insect cyborg control. To this end, we measured joint torques induced by stimulating one out of three leg muscles in the middle leg of the stick insect species *Carausius morosus* (*Sinéty, 1901*): these were the protractor coxae, retractor coxae, and levator trochanteris. We focused on these three proximal muscles because the retractor coxae is the primary muscle for propulsion (*Rosenbaum et al., 2010*), the pro/retractor coxae contributes to weight-dependent postural adjustment by regulating joint stiffness (*Dallmann et al., 2019*; *Günzel et al., 2022*), and the levator trochanteris is important for postural termination and swing initiation (*Dallmann et al., 2017*). Using a custom-built electrical stimulator to generate parameter-tunable pulse-width-modulated (PWM) signals, we simulated burst-like activity of motor neurons in insects and measured the corresponding joint torques generated in response to our electrical stimuli. Using Bayesian statistical modeling and the 'widely applied information criterion' (WAIC) index (*Watanabe, 2018*) for model prediction, we evaluated six model variants, namely a simple linear, hierarchical linear, simple nonlinear, and three hierarchical nonlinear models, to identify the model that best explained the experimental data. In particular, we evaluated the predictive performances of model variants with and without inter-individual variation of experimental parameter estimates. A piecewise linear relationship was observed between the burst duration and the joint torque generated for a given parameter set of the PWM burst. Linearity was found to hold for burst durations of up to 500ms, which corresponds to the stance phase (300–500ms) and swing phase (to 250ms) of a stick insect walking at medium to fast speeds (*Dürr et al., 2018*). Furthermore, the hierarchical Bayesian modeling revealed both invariant and individually varying characteristics of joint torque generation in stick insects. This allows for individual tuning of electrical stimulation parameters for highly precise insect cyborg control.

With regard to our general understanding of insect motor control, our study demonstrates that the dependency of joint torque on electrical stimulus duration is linear, despite nonlinear activation and contraction dynamics of insect muscle. Furthermore, the proposed hierarchical Bayesian model allows for a quick, simple and reliable measurement of the individual characteristics and, therefore, quantification of inter-individual differences. Whereas several studies have reported on inter-individual differences in neural (*Golowasch et al., 2002*; *Schulz et al., 2006*) and muscle activity (*Horn et al., 2004*; *Brezina et al., 2005*; *Zhurov et al., 2005*; *Blümel et al., 2012b*; *Blümel et al., 2012a*; *Thuma et al., 2003*; *Hooper et al., 2006*), we propose how hierarchical Bayesian models may be used to harness inter-individual differences in insect locomotion research.

## Results

### Joint torque measurements

Since movement at a given leg joint is effected by joint torque, the goal of our experimental measurements was measure jont torque as a function of electrical stimulation. This was done for the two proximal joints of the stick insect middle leg. The insect was fixed dorsal side up on a wooden support, with its right middle leg coxa reaching beyond the edge (*Figure 1A* right). We selected three leg muscles (protractor, retractor, and levator) for electrical stimulation (*Figure 1B*). When stick insects walk, they use the protractor to swing the leg forward during the swing phase, the retractor to move the leg backward during the stance phase, and levator to initiate the stance-to-swing transition (*Rosenbaum et al., 2010*; *Dallmann et al., 2019*; *Günzel et al., 2022*; *Bässler and Wegner, 1983*). Moreover, co-contraction of the protractor and retractor are known to vary based on the overall load distribution, thus being important for postural control by regulating joint stiffness (*Dallmann et al., 2019*; *Günzel et al., 2022*). Accordingly, electrical stimulation of the protractor and retractor muscles generated forward and backward movements at the thorax–coxa (ThC) joint, as measured by a calibrated custom-made force transducer with a strain gauge held against the femur with known distance from

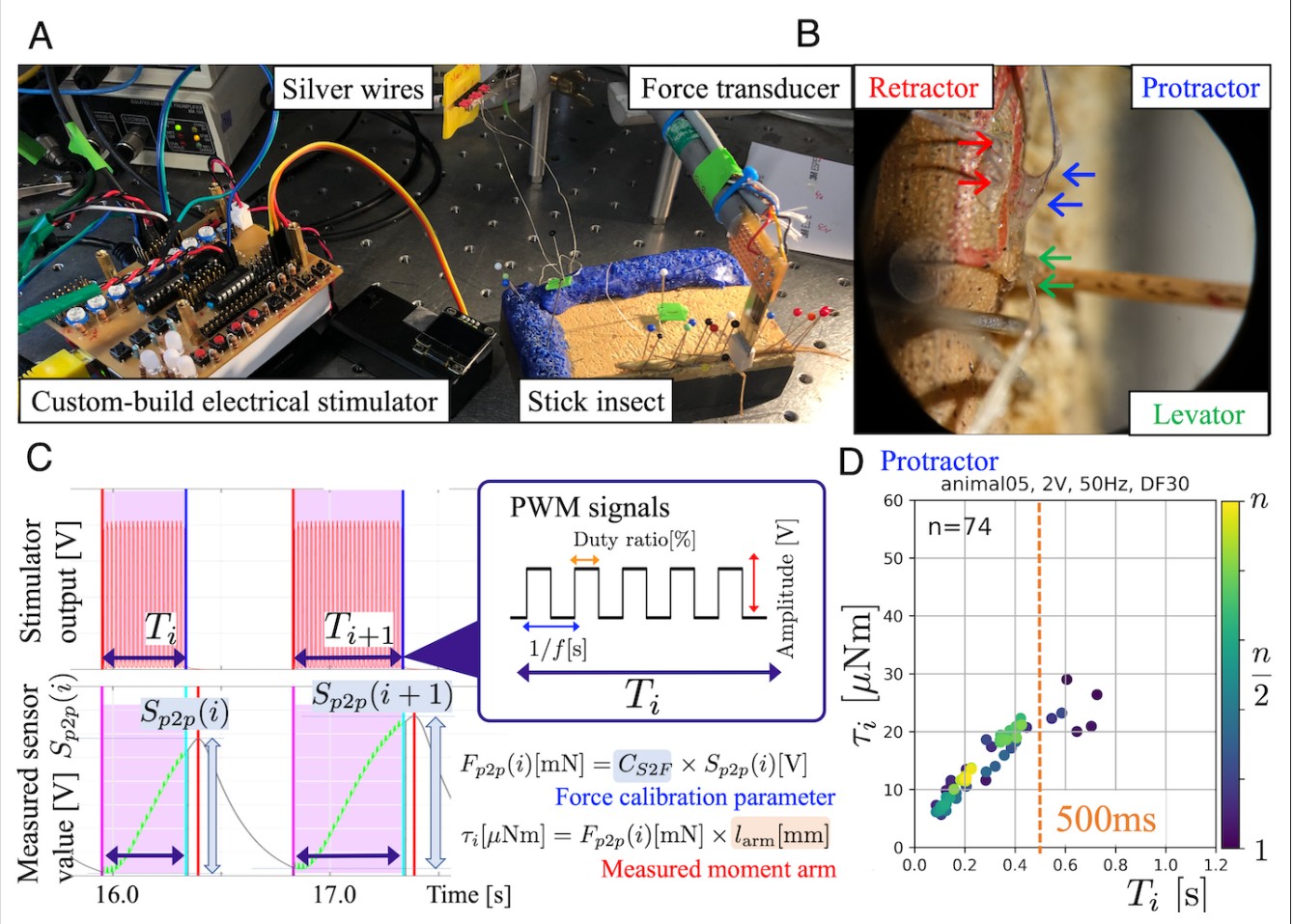

**Figure 1.** Experimental setup and joint torque calculation (**A**) The insect was fixed dorsal side up on a balsa wood platform. Two small insect pins attached to the tip of the force transducer held the middle part of the femur segment of the middle leg. (**B**) Electrodes (arrows) implanted into the three leg muscles protractor, retractor, and levator, in the right middle leg. (**C**) We systematically analyzed how joint torques depended on the three PWM burst parameters amplitude [V], frequency [Hz], and duty ratio [%], and identified the combinations that most effectively and repeatedly produced torque. The upper graph shows the profile of an electrical stimulation signal for each muscle. The lower graph shows the profile of the sensor value measured with the force transducer. (**D**). The panel shows the calculated ThC-joint torque profile versus the burst duration $T_i$ during the protractor stimulations (animal05, $n = 74$). In this experiment, the burst duration $T_i$ was varied at random, and the torque was calculated from force measurements with calibrated conversion factor and moment arm (see (**C**)). The voltage, frequency, and duty ratio of the PWM signals were 2.0 V, 50 Hz, and 30%, respectively. The color of the dots represents the number of stimulations (blue–yellow: 1–74). The orange dotted vertical line indicates $T_i$ at 500ms.

the joint. Stimulation of the levator muscle generated an upward movement at the coxa–trochanter (CTr) joint (*Dallmann et al., 2016*).

*Figure 2* illustrates the obtained relation between the PWM burst duration and the generated joint torques for the protractor (A), retractor (B), and levator (C) muscles from 10 animals ($N = 10$). The parameters of the PWM signals were set to 2.0 V, 50 Hz, and 30% duty ratio. During one trial, we stimulated one muscle $n$ times with fixed PWM parameters and measured the generated torque at the corresponding joint.

The results indicate the input–output relation (burst duration and generated torque) corresponded to a linear function or a power function with an exponent of less than 1.0. Furthermore, the relationship holds for burst durations up to 500ms for all animals, corresponding to the duration of swing and stance phases in medium to fast walking stick insects (*Dürr et al., 2018*). Maximum torques for ThC and CTr joints were 60 μNm, 120 μNm, and 40 μNm (*Dallmann et al., 2019*). For a given set of PWM parameters, the generated torque characteristics remained almost constant during all stimulations under the same condition, suggesting that muscle fatigue or warm-up effects were negligible for

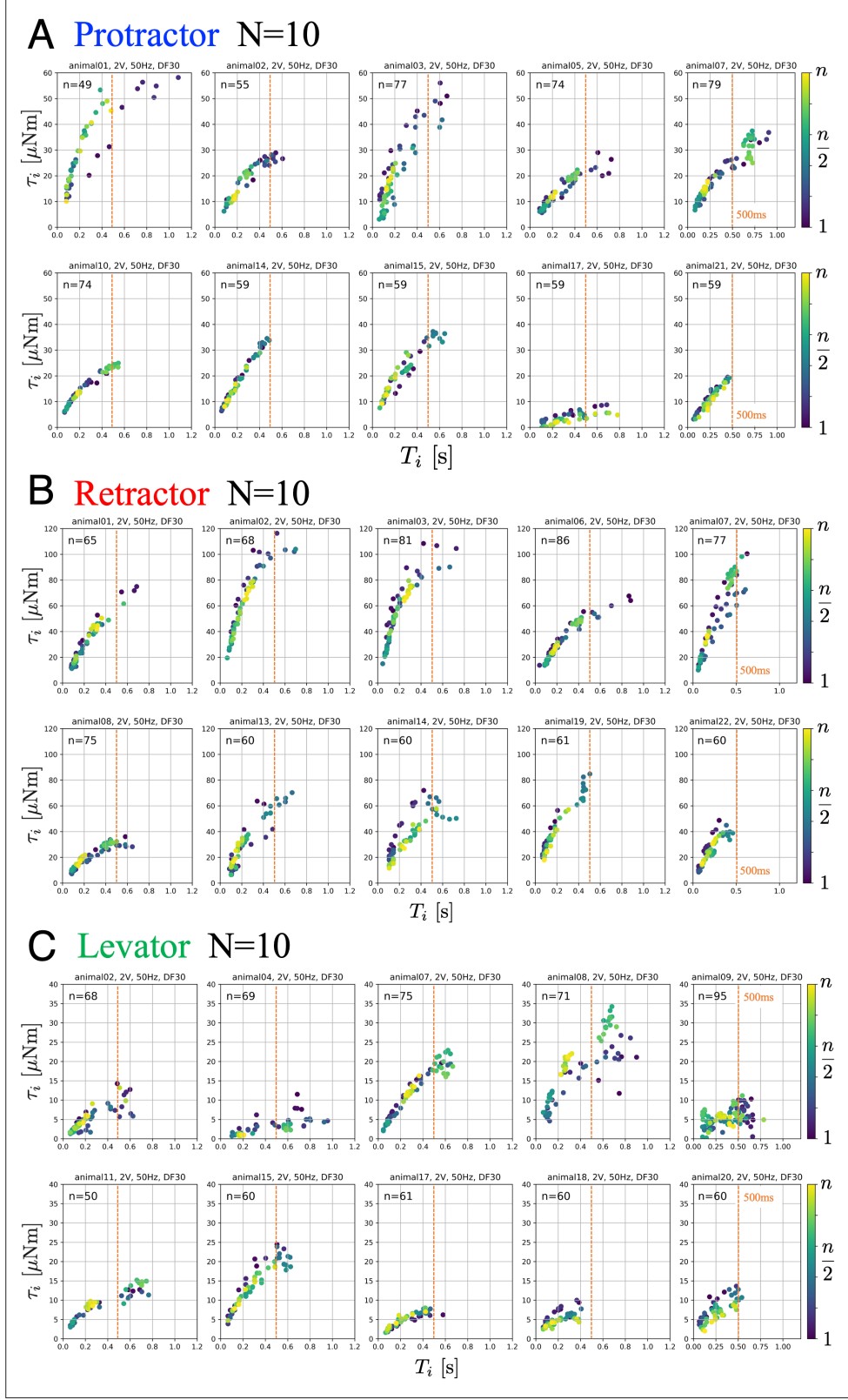

**Figure 2.** Joint torques as a function of burst duration. Data from 10 animals with three muscles, each: (**A**) protractor, (**B**) retractor, and (**C**) levator muscle stimulation. The PWM burst parameters were 2.0 V. 50 Hz, and 30% duty ratio. $n$ gives the number of stimulations for each animal. Electrical stimulations were performed manually and randomly; therefore, the total number of stimuli was different for each animal. The color of the

*Figure 2 continued on next page*

*Figure 2 continued*

symbols indicates the order of the stimulations: blue (1) to yellow (*n*). The positive values of joint torque represent intended (**A**) forward, (**B**) backward and (**C**) upward rotation of the coxa relative to the thorax. Source code and data are available on Dryad (Figure2.zip, https://doi.org/10.5061/dryad.wpzgmsbsw).

at least $n = 50$ stimulations. Furthermore, we verified that no significant changes occurred in muscle characteristics owing to the pre- and post-experimental relationship.

## Bayesian statistical modeling

To investigate joint-torque properties generated by muscle stimulation while explicitly considering inter-individual variation of muscle physiology, we used a Bayesian statistical analysis and modeling framework. The probabilistic nature of Bayesian models makes them appropriate for modeling 'uncertainty', as introduced by inter-individual variation (*Gelman et al., 2013*). Bayesian analysis can be used to estimate a probabilistic distribution (model) that encodes an unknown observation target by using observed data and updating the distribution in the model. Furthermore, hierarchical model variants allow the inclusion of a hyperparameter, thus allowing for a parameter of choice to be drawn from yet another probabilistic distribution. In our case, hierarchical-model variants were used to account for inter-individual differences (*Watanabe, 2018*).

Here, we modeled the relationship between the burst duration of the electrical stimulation and the joint torque generated using a single model (a power function) with six variants (for details, see subsection 'Models'). All model variants were specified in a probabilistic programming language developed by Stan (*Stan Development Team, 2023*). Here, we used non-informative uniform priors for the parameters $\beta$, $\gamma$, and $\sigma$, unless stated otherwise. For estimation, we used the numerical Markov Chain Monte Carlo (MCMC) method, and scripted the models in R (v.4.1.3; *R Development Core Team, 2023*), in which the Stan code was compiled and executed using the R package 'rstan' (*Stan Development Team, 2023*). The software performed sampling from prior distributions using No-U-Turn Sampler (NUTS; *Hoffman and Gelman, 2014*). Sampling convergence was detected through trace plots and the quantitative Gelman–Rubin convergence statistic $R_{hat}$ (*Gelman and Rubin, 1992*), where $R_{hat} < 1.10$.

## Models

$\tau_i$ and $T_i$ represent the calculated joint torque based on the force-transducer value and the burst duration of a PWM signal for electrical stimulation, respectively. We assumed that $\tau_i$ follows a normal distribution, described by the $\mathcal{N}(\mu, \sigma)$ function, where μ and $\sigma$ represent the mean and standard deviation (S.D.) of the distribution. Indexes $i$ and $j$ represent the numbers of stimulations and animals, respectively.

Model 1-1: Linear model representing the linear relationship between burst duration and joint torque

$$\tau_i \sim \mathcal{N}(\mu = \beta T_i, \sigma), \tag{1}$$

where $\beta$ represents the inclination of the estimated linear function.

Model 1-2: Hierarchical model representing the linear relation between burst duration and joint torque

$$\tau_{i,j} \sim \mathcal{N}(\mu = \beta_j T_i, \sigma), \tag{2}$$

$$\beta_j \sim \mathcal{N}(\mu = \mu_\beta, \sigma_\beta), \tag{3}$$

where $\beta_j$ represents the inclination of the estimated linear function on the ($j$)th animal. Furthermore, in this hierarchical model, $\beta_j$ is drawn out of a normal distribution that captures inter-individual variation, where $\mu_\beta$ and $\sigma_\beta$ represent the mean and S.D. of the distribution, respectively.

Model 2-1: Non-linear model representing the nonlinear relationship between burst duration and joint torque

$$\tau_i \sim \mathcal{N}(\mu = \beta \{T_i\}^\gamma, \sigma), \tag{4}$$

where, $\beta$ and $\gamma$ represent the magnitude of the base and exponent of the estimated non-linear power function, respectively.

Model 2-2: Hierarchical model representing the nonlinear relationship between burst duration and joint torque

$$\tau_{i,j} \sim \mathcal{N}(\mu = \beta_j \{T_i\}^\gamma, \sigma), \tag{5}$$

$$\beta_j \sim \mathcal{N}(\mu = \mu_\beta, \sigma_\beta), \tag{6}$$

where $\beta_j$ and $\gamma$ represent the magnitude of the base on the ($j$)th animal and the exponent of the estimated nonlinear power function, respectively. In this model, $\beta_j$ follows a normal distribution as described above, where, $\mu_\beta$ and $\sigma_\beta$ represent the mean and S.D. of the distribution.

Model 2-3: Hierarchical model representing the nonlinear relation between burst duration and joint torque

$$\tau_{i,j} \sim \mathcal{N}(\mu = \beta \{T_i\}^{\gamma_j}, \sigma), \tag{7}$$

$$\gamma_j \sim \mathcal{N}(\mu = \mu_\gamma, \sigma_\gamma), \tag{8}$$

where $\beta$ and $\gamma_j$ represent the magnitude of the base and the exponent on the ($j$)th animal for the estimated nonlinear, power function, respectively. In this model, $\gamma_j$ follows a normal distribution as described above, where $\mu_\gamma$ and $\sigma_\gamma$ represent the mean and S.D. of the distribution.

Model 2-4: Hierarchical model representing the nonlinear relationship between burst duration and joint torque

$$\tau_{i,j} \sim \mathcal{N}(\mu = \beta_j \{T_i\}^{\gamma_j}, \sigma), \tag{9}$$

$$\beta_j \sim \mathcal{N}(\mu = \mu_\beta, \sigma_\beta), \tag{10}$$

$$\gamma_j \sim \mathcal{N}(\mu = \mu_\gamma, \sigma_\gamma), \tag{11}$$

where $\beta_j$ and $\gamma_j$ represent the magnitude of the base and the exponent of the estimated nonlinear power function, respectively, on the ($j$)th animal. In this model, $\beta_j$ and $\gamma_j$ follow normal distributions as described above, where $\mu_\beta$ and $\mu_\beta$ are the means, and $\sigma_\gamma$ and $\sigma_\beta$ are the S.D.s of the distribution.

## Comparison of model predictability

Using the WAIC described in the 'Materials and methods' section, we compared the prediction performance of the six models. *Figure 3 (A)–(C)* shows the WAIC values for each voltage applied (1.0–4.0 V) and models 1–1 to 2–4. *Figure 3 (A)–(C)* show the results for the protractor, retractor, and levator muscles, respectively. For stimulation experiments on each of the three muscles, the models with a hierarchical parameter for expressing individual differences for $\tilde{\beta}$ (models 1–2 and 2–2) had the lowest WAIC and, therefore, the best predictive performance. Conversely, the model with individual differences for both $\tilde{\beta}$ and $\tilde{\gamma}$ (model 2–4) exhibited the lowest prediction performance, indicating that inter-individual variation of the exponent does not improve model estimates.

## Bayesian estimation of the generated torque for a given burst duration

*Figure 3(D)-(F)* shows the predictive distributions for data of a new animal using the Bayesian posterior distribution for the six models. The results were obtained with PWM bursts at 2.0 V voltage, 50 Hz frequency, and 30% duty ratio. The results show that the hierarchical models (model 1–2 and model 2–2) for the $\tilde{\beta}$ parameter can successfully and adequately capture the range of experimental results on (D) protractor, (E) retractor, and (F) levator torques for all animals. This suggests that, compared with other models, the hierarchical models can appropriately account for inter-individual variation of muscle properties for new unknown animals. *Figure 4* depicts the distributions predicted by the linear hierarchical model (model 1–2) for each individual by overlapping the experimental data shown in *Figure 2*.

## Effect of an individual animal and applied voltage on muscle properties

*Figure 5* presents the variations in the muscle characteristic parameters $\beta$ and $\gamma$ in response to changes in the applied voltage. In the voltage-change experiments, we followed a specific order of voltage application, gradually increasing from 1 V to 4 V, for each individual. Furthermore, we confirmed that

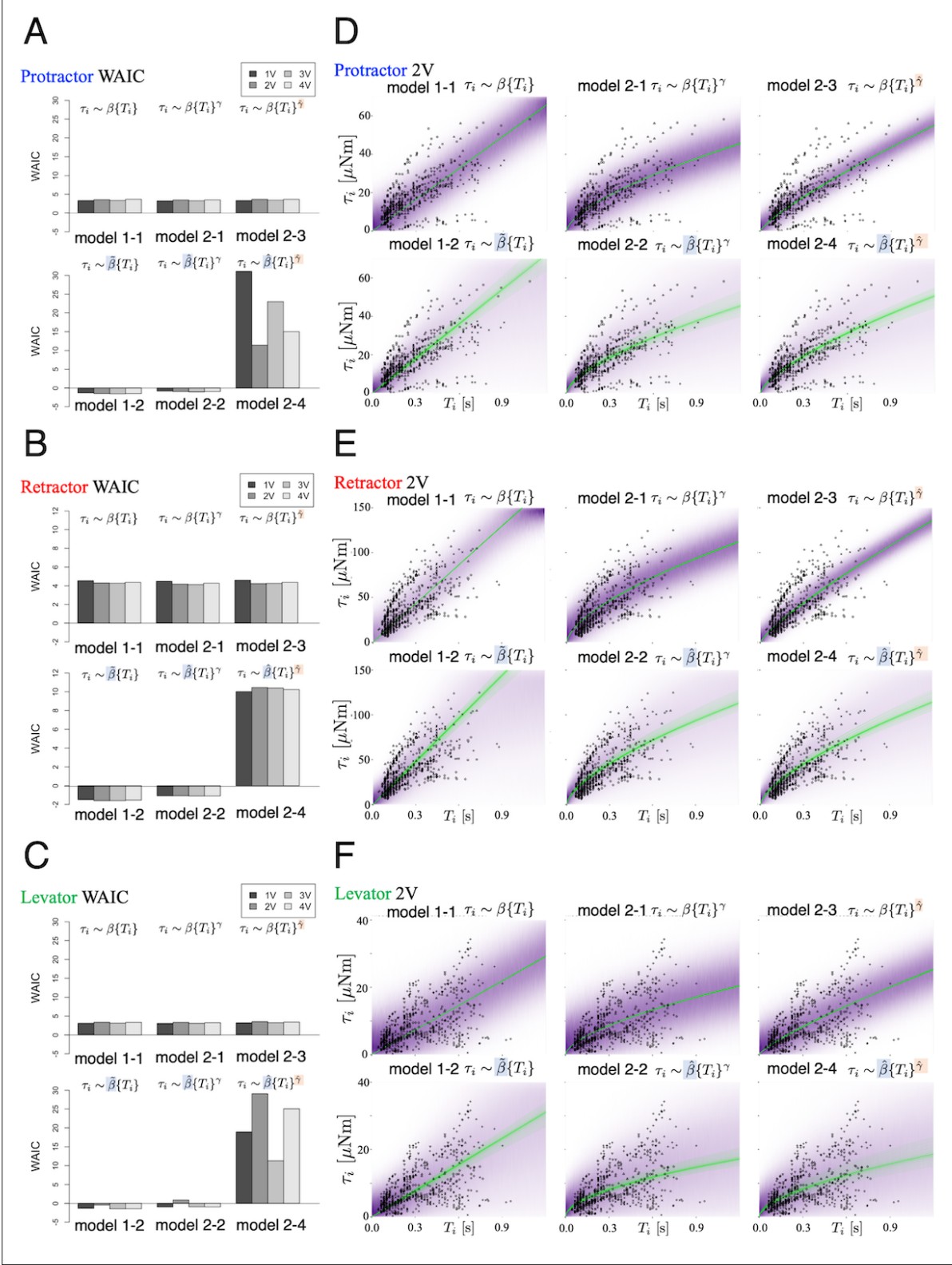

**Figure 3.** Model comparison underscores significance of inter-individual variation of slope. We compared the six models that were explained in the 'Model' subsection. (**A**), (**B**), and (**C**) show plots of the WAIC (*Watanabe, 2018*) values for the protractor, retractor, and levator stimulations, respectively ($N = 10$ animals per muscle). The parameters with tilde, $\tilde{\beta}$ and $\tilde{\gamma}$, indicate that the parameters include inter-individual variation. PWM parameters were set as follows: (1.0 V, 2.0 V, 3.0 V, and 4.0 V), at 50 Hz and 30% duty ratio. Negative values were obtained for models 1–2 and 2–2 for all voltages

*Figure 3 continued on next page*

*Figure 3 continued*

and all muscles. The lowest WAIC indicates the best prediction model, as explained in the "WAIC" subsection. Right panels show Bayesian predictive estimation for the protractor (**D**), retractor (**E**), and levator (**F**) stimulation experiments with PWM parameters 2.0 V, 50 Hz, and 30% duty ratio. The differences in the point styles indicate individual animals. In each panel, the violet shading indicates the probability density of the distribution predictive. The green lines represent twenty samples from the posterior distribution in decreasing order of probability density. Source code and data are available on Dryad (Figure3-5.zip, https://doi.org/10.5061/dryad.wpzgmsbsw).

applying voltages ranging between 1 and 4 V did not induce fatigue. *Table 1* summarizes the number of electrical stimuli administered to each muscle in each individual. We determined the changes in $\beta$ and $\gamma$ with respect to the applied voltage by analyzing the experimental results using the six Bayesian models.

The results indicate the following three points: (1) $\beta$ varied with the applied voltage, and there exists an optimal voltage that maximizes $\beta$; (2) except for non-hierarchical nonlinear models (models 2–1 and 2–3), $\gamma$ has a low dependence on the applied voltage; and (3) $\beta$ is strongly subject to inter-individual variation (large variability), whereas $\gamma$ is affected much less.

## Discussion

In this study, we investigated externally controlled joint torques induced by external electrical stimulation of one out of three leg muscles (protractor, retractor, and levator) in the stick insect *Carausius morosus*. For a given parameter set for PWM burst stimulation, we found a piecewise linear relationship between the burst duration and generated joint torque. Linearity holds for a burst duration up to 500ms. For a more detailed analysis of the joint torques generated by leg muscles, we used Bayesian statistical analysis and modeling to account for inter-individual variation. A comparison of the six models (with combinations of linear, nonlinear, non-hierarchical, and hierarchical models) showed that the two models that include inter-individual variation of slope parameter $\beta$ performed best. Models 1–2 and 2–2 provide the most accurate predictions of the posterior predictive distribution.

The exponent $\gamma$ is a macroscopic property of the generated joint torque, that is the degree of non-linearity of the stimulus-torque characteristic; it is linear when $\gamma = 1$. Conversely, slope parameter $\beta$ defines the rate of increase of the generated torque. In a comparison of the prediction performance of models in *Figure 3*, the mathematical index WAIC revealed that the models 1–2 and 2–2, wherein only $\beta$ was a hierarchical parameter, performed the best. Since only hierarchical parameters account for inter-individual variation, we conclude that $\beta$ is strongly affected by individual differences, whereas $\gamma$ is invariant among specimens. Thus, we found that the macroscopic properties of leg muscles are common to all individuals, whereas individuals differ in the slope $\beta$, that is the rate by which the three types of leg muscles respond to electrical stimulation. Furthermore, as shown in *Figure 5*, we found that $\beta$ was highly affected by the applied voltage, whereas the exponent $\gamma$ was close to unity, largely independent of the applied voltage, indicating that the macroscopic properties of leg muscles were invariant to the applied voltage. We conclude that linearity was an invariant feature of the stimulus-torque characteristic, whereas the slope of this characteristic varies among individual stick insects and with the applied voltage. These results are in line with those of existing studies on the properties of myogenic forces in other insect species (*Cao et al., 2014*; *Blümel et al., 2012a*; *Harischandra et al., 2019*): The generated torque depends considerably less on the PWM voltage and frequency (*Blümel et al., 2012a*; *Harischandra et al., 2019*) than on the burst duration, suggesting that the total number of subsequent input pulses is important. This is indeed expected for a slow insect muscle (*Blümel et al., 2012a*) that essentially 'counts' incoming spikes within a given time window. Compared to the nonlinear properties of muscle, we demonstrated that our monitoring of torques in an intact animal resulted in a linear characteristic (for intervals up to 500ms) that would not be expected from isometric force measurements of isolated muscles. Furthermore, changing the PWM frequency was found to be comparable to changing the number of spikes over a given period, whereas changing the duty ratio was found to be comparable to varying the average voltage over a given period (see *Appendix 1— figures 1 and 2*). Therefore, from both technical and cyborg control viewpoints, the control of burst duration provides beneficial insights into feasibility.

The comparison of the linear model (model 1–2) with the nonlinear model (model 2–2) using the WAIC for all conditions (muscle type and applied voltage) resulted in lower values for the linear

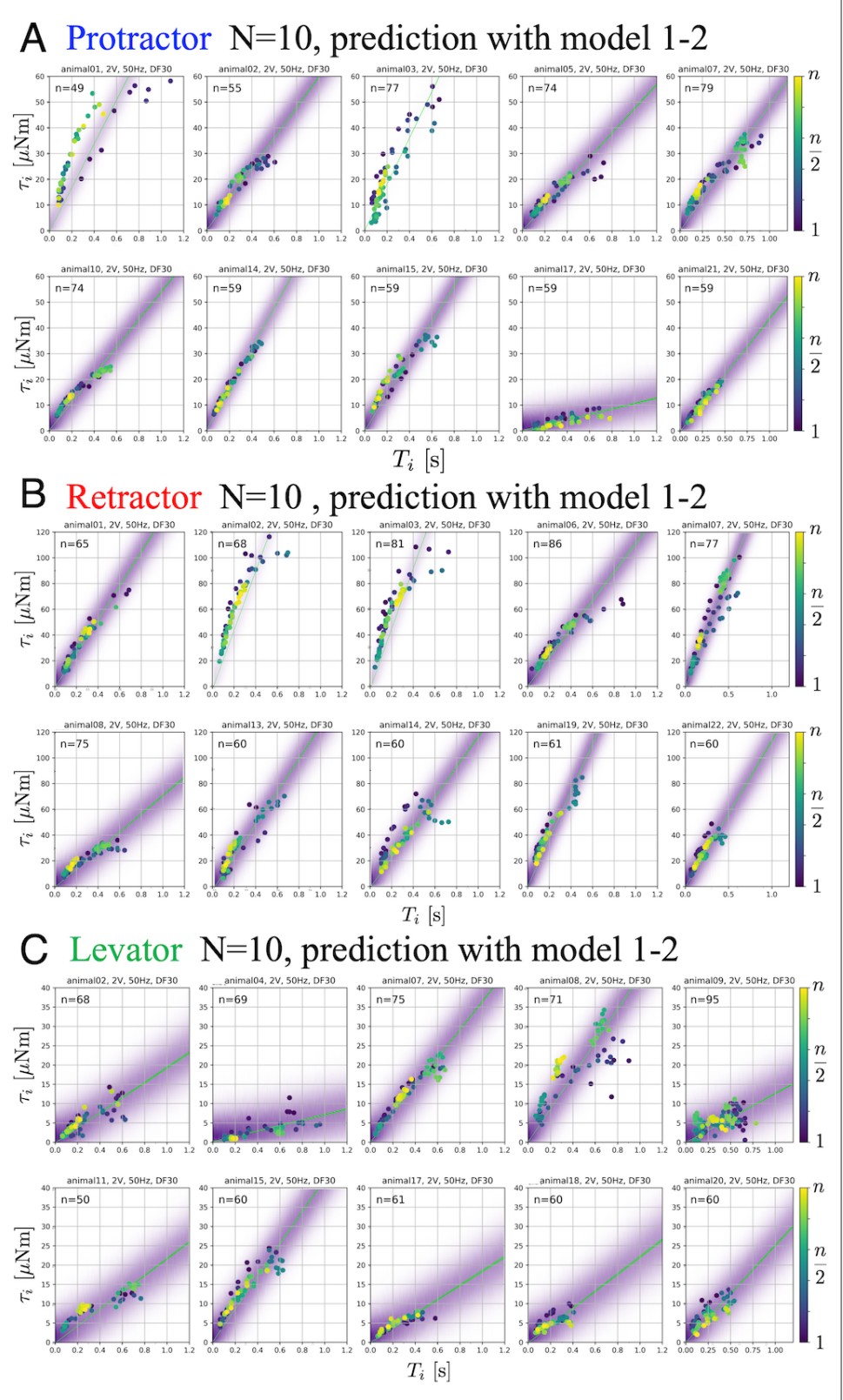

**Figure 4.** Predictive distributions from the linear hierarchical model (1-2) for each individual: The protractor (**A**), retractor (**B**), and levator (**C**) stimulation experiments with PWM parameters, 2.0 V, 50 Hz, and 30% duty ratio. $n$ gives the number of stimulations for each animal. The color legend indicates the order of the stimulations: blue (1) to yellow ($n$). In each panel, the violet shading indicates the probability density of the predictive distribution. The

*Figure 4 continued on next page*

*Figure 4 continued*

green lines represent twenty samples from the posterior distribution in decreasing order of probability density. The results demonstrate that the linear hierarchical model had an accurate predictive distribution in the range up to 500ms. Source code and data are available on Dryad (Figure3-5.zip, https://doi.org/10.5061/dryad.wpzgmsbsw).

---

model. Models with lower WAIC can generate predictive distributions closer to the true distribution while using fewer parameters (*Watanabe, 2018*), suggesting that the experimental results obtained in this study can be adequately explained using a linear hierarchical Bayesian model (1-2). This model renders it useful for predicting the generated torque for each new animal in real-time during an experiment. Specifically, by assuming the linear hierarchical Bayesian model, we can measure responses to very few PWM stimulus bursts and estimate $\beta$ for the current individual's stimulus-torque characteristic. This allows an experimenter to acquire an appropriate muscle model of an unknown animal in a short time without having to use potentially time-consuming machine learning methods, such as deep learning algorithms. Moreover, the properties were linear for stimulus burst durations up to 500ms. This linearity region corresponds to the stance and swing phase durations of medium-speed to fast-walking stick insects of the species *Carausius morosus* (*Dürr et al., 2018*). The magnitudes of the joint torques generated by the protractor, retractor, and levator were comparable to those for resisted movement during stick-insect walking, for example coxa-trochanter joint depression during stance (*Dallmann et al., 2016*). This suggests that the estimated stimulus-torque characteristic captures the natural dynamic properties of leg muscles during walking in terms of both the duration of excitation and maximum torque. However, the hierarchical nonlinear model (model 2–2) would be more appropriate for estimating properties related to longer time scales, such as those associated with the complete range of muscle excitation. Nevertheless, we emphasize once again that a key contribution of this study lies in demonstrating, based on experimental data, that the muscle property $\gamma$ across the complete excitation range exhibits inter-individual variations and is independent of linear or nonlinear properties; hence, the weight $\hat{\beta}$ assigned to these properties represents individual differences.

This study takes a first but important step towards highly precise insect cyborg control. In previous studies, we defined Motion Hacking (*Owaki et al., 2019*; *Owaki and Dürr, 2022*) as a technique for controlling insect leg movements through external electrical stimulation, while retaining the insect's own nervous system and sensorimotor loops. This approach requires a collaborative effort of engineering and biology in order to elucidate how adaptive walking ability of insects may be exploited for biohybrid control of motor flexibility. The Motion Hacking (*Owaki et al., 2019*; *Owaki and Dürr, 2022*) method strives to observe the adaptation process in the insect's own sensorimotor system as leg movements are intentionally controlled by a human operator, so as to reveal hidden mechanisms underlying natural locomotion. Thus far, research on insect cyborg control has addressed aspects of flight control (*Sato, 2009*; *Sato and Maharbiz, 2010*; *Sato et al., 2015*; *Kosaka et al., 2021*; *Sane et al., 2007*; *Bozkurt et al., 2009*; *Hinterwirth et al., 2012*), gait control (*Cao et al., 2016*; *Vo Doan et al., 2018*; *Nguyen et al., 2020*; *Ando and Kanzaki, 2017*; *Sanchez et al., 2015*), and controlling jellyfish propulsion (*Xu and Dabiri, 2020a*; *Xu et al., 2020b*; *Xu et al., 2020c*). In contrast to our present study, the main objective of the mentioned studies was to convert target animals into cyborgs, with little examination of the control mechanisms and/or muscle properties involved. Here, we used PWM pulse bursts to mimic motor neuron commands during insect locomotion, and selected key muscles to estimate stimulus-torque characteristics reliably and in very short time. Then, we used Bayesian statistical modeling to tell which parameters were subject to inter-individual variation and which were not. Our finding of linear characteristics with inter-individual variation of slope show compellingly how a systematic engineering intervention to an otherwise intact animal motor system can yield a simple, technically exploitable description of motor system properties. We argue that this description could not have been obtained by methods addressing isolated neural circuits or partial anatomical structures, but required the physical intactness of the natural system.

The contributions of this research are as follows: (1) this study demonstrates that compared to the nonlinear activation and contraction dynamics of insect muscles, the joint torque generated through electrical stimulation increases linearly with the duration of the stimulus, particularly during the stance and swing phases that characterize stick insect locomotion; (2) it introduces a hierarchical Bayesian model that allows for a reliable and simple description of the individual differences observed in neuromuscular system parameters. These contributions not only advance the field of insect cyborg control

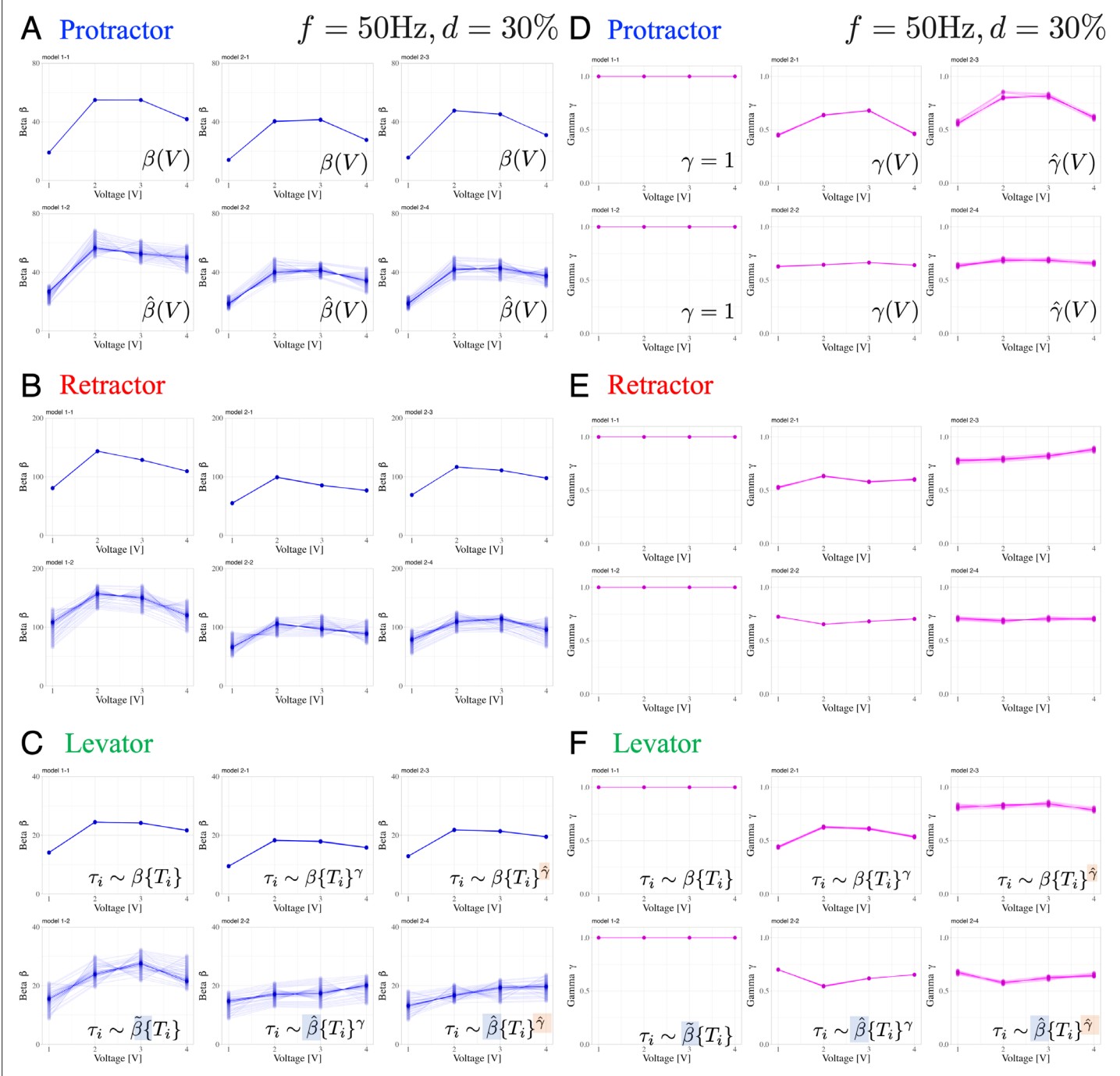

**Figure 5.** Dependence of muscle property parameters on the applied voltage and individual animals in the six models: The left graphs (**A**), (**B**), and (**C**) represent the estimation for $\beta$ for the applied voltage varied from 1.0 to 4.0 V for the six models. (**A**) and (**D**), (**B**) and (**E**), and (**C**) and (**F**) illustrate the protractor, retractor, and levator stimulations, respectively. In (**A**) to (**C**), the upper and lower panels show non-hierarchical (1-1, 2-1, 2-3) and hierarchical (1-2, 2-2, 2-4) models for $\beta$, respectively. The right graphs (**D**), (**E**), and (**F**) represent the estimation for $\gamma$ in applied voltage changes. In (**D**) to (**F**), the left panel shows linear models ($\gamma = 1$, 1–1, 1–2); the middle and right panels illustrate non-hierarchical (2-1, 2-2), and hierarchical (2-3, 2-4) models for $\gamma$, respectively. For hierarchical models (1-2, 2-2, 2-3, 2-4), the plot includes thirty samples from the posterior distribution in decreasing order of probability density, showing inter-individual variation. Source code and data are available on Dryad (Figure3-5.zip, https://doi.org/10.5061/dryad.wpzgmsbsw).

**Table 1.** List of Stick insects used in the stimulation experiments for each muscle.

Animal ** denotes the identification number of the stick insects. We analyzed 20 animal data from 'Animal 01' to 'Animal 22' except for '12' and '16'. Due to experimental failures and time limitations, we could not obtain stimulation data for all three muscles from the same animal on the same day. Therefore, we conducted experiments to collect data for ten animals ($N = 10$) for each muscle through experiments using 20 animals.

| Date | Protractor | 1,2,3,4 V | Retractor | 1,2,3,4 V | Levator | 1,2,3,4 V |
|------|-----------|-----------|-----------|-----------|---------|-----------|
| 2018.8.21. | Animal 01 | 50,49,60,50 | Animal 01 | 60,65,55,72 | | |
| 2018.8.22. | Animal 02 | 49,55,51,40 | Animal 02 | 55,68,58,65 | Animal 02 | 37,68,67,70 |
| 2018.8.27. | Animal 03 | 60,77,91,72 | Animal 03 | 94,81,63,75 | | |
| 2018.8.28. | | | | | Animal 04 | 74,69,80,77 |
| 2018.8.30. | Animal 05 | 67,74,79,79 | | | | |
| 2018.8.31. | | | Animal 06 | 75,86,76,76 | | |
| 2018.9.03. | Animal 07 | 81,79,74,78 | Animal 07 | 82,77,81,75 | Animal 07 | 75,75,75,75 |
| 2018.9.04. | | | Animal 08 | 75,75,79,75 | Animal 08 | 75,71,77,84 |
| 2018.9.05. | Animal 10 | 74,74,74,75 | | | Animal 09 | 75,95,75,75 |
| 2018.9.19. | | | Animal 13 | 62,60,70,69 | Animal 11 | 50,50,50,50 |
| 2018.9.20. | Animal 14 | 59,59,60,59 | Animal 14 | 60,60,60,60 | | |
| 2018.9.21. | Animal 15 | 59,59,59,59 | | | Animal 15 | 61,60,60,51 |
| 2018.9.23. | Animal 17 | 59,59,60,69 | | | Animal 17 | 59,61,60,61 |
| 2018.9.24. | | | Animal 19 | 61,61,61,60 | Animal 18 | 60,60,59,66 |
| 2018.9.25. | | | | | Animal 20 | 62,60,70,60 |
| 2018.9.26. | Animal 21 | 59,59,60,59 | Animal 22 | 60,60,60,60 | | |
| Total | N=10 | | N=10 | | N=10 | |

but also enhance our understanding of insect locomotion mechanisms. Animal locomotion is not solely governed by the brain and nervous system but also relies on the physical properties of the body and its interactions with the environment (*Chiel et al., 2009*; *Nishikawa et al., 2007*). Arthropods, in particular, effectively utilize mechanical properties and environmental interactions in their locomotion. Studies have revealed various related strategies, including the joint stiffness nonlinearity and hysteresis in spiders (*Blickhan, 1986*), generation of large motor outputs during escape maneuvers (*Card, 2012*) and posture stabilization (*Blickhan, 1986*) by adjusting the joint stiffness, mechanical sensing based on frequency characteristics that vary with the joint stiffness and posture in web-making spiders (*Blickhan and Barth, 1985*; *Mhatre et al., 2018*), transitions in movement patterns in response to mechanical interactions with the environment (*Othayoth et al., 2020*), and transitions in the coordinated movements of the body and legs (*Wang et al., 2022*) in cockroaches. Similarly, to elucidate the animal locomotion mechanisms emerging from such complex interactions, *Sponberg et al., 2011a*; *Sponberg et al., 2011b* conducted experiments similar to those in our study by perturbing neural feedback through artificial interventions on muscle action potentials (MAPs) in cockroaches *Blaberus discoidalis (L.)*. In follow-up studies, we further estimated the passive joint stiffness and analyzed the phase responses of stick insects during walking by accurately controlling the joint torque based on the linear stimulus duration-joint torque model derived in this study. We believe that these approaches will contribute to a deeper understanding of stick insect walking mechanisms, such as their use of two different stride lengths in response to their environment (*Theunissen and Dürr, 2013*).

Still, there are several limitations to the present study. First, as in many neurophysiological experiments (*Berg et al., 2012*; *Lepreux et al., 2019*), stick insects were fixed and not walking in the experimental setup (*Figure 1A*). Although there are only few studies on the natural dispersal behavior of stick insects, it is clear that they spend a lot of their lifetime at rest, for example in camouflage.

Their tendency to attain camouflage postures can be exploited in experiments, as it is relatively easy to restrain active, spontaneous leg movements in an experimental setup. Nevertheless, the possibility to conduct combined motion capture and EMG recordings in freely walking stick insects (*Dallmann et al., 2019*; *Günzel et al., 2022*; *Dallmann et al., 2017*; *Bidaye et al., 2018*) suggests that Motion Hacking during unrestrained, voluntary locomotion will be feasible in the future. Whereas the range of PWM burst duration and the joint torques generated are well within the physiological range, there is still considerable discrepancy between the PWM signals generated by our Raspberry Pi microcontroller and the natural firing patterns of motor neuron pools (*Günzel et al., 2022*). Future research will need to examine how much the simplification of the driving burst input affects the time course of the torque generated. So far, it is re-assuring that the simplified PWM signal used here could be applied more than 50 times in a sequence without causing muscle fatigue, that is with a sustained level of generated torque.

Finally, so far we have not fully investigated the effects of the electrical muscle stimulation on sensory feedback. The maximum voltage of 4 V used here did not cause abnormal motion that could be attributed to cross-talk stimulation of sensory afferents. Therefore, we conclude that unintended electrical stimulation of sensory afferents was negligible. Moreover, control measurements confirmed that muscles other than those stimulated by the electrodes were not active and did not generate force, as it would be expected from unintended stimulation via cross-talk. More generally, the activation of sensory organs during cyborg control is an interesting topic, with strong potential for expanding the concept of Motion Hacking. In the future, we will examine the performance of external leg movement control in an experimental setup, both without load (i.e. on a tether, without substrate contact) and with natural load distribution (i.e. by intervention during free walking). We are confident that these experiments, will provide further support of the Motion Hacking method and will reveal findings that could not be obtained by more conventional experiments without external stimulation of the neuro-muscular system. This will also contribute to potential applications in highly precise insect cyborg control.

## Materials and methods

### Animals

We tested 20 adult female *Carausius morosus* from our laboratory colony at Bielefeld University in 2018. The animals were raised under a 12 hr:12 hr light:dark cycle at a temperature of 23.9 ±1.3 °C (mean ± S.D). All experiments were conducted at room temperature (20–24 °C). *Table 1* lists the stick insects used in the electrical stimulation experiments. Owing to a combination of experimental failures and time constraints, we could not obtain stimulation data for all three muscles from the same animal on a single day. Therefore, we collected data from 10 animals ($N = 10$) for each muscle through experiments with 20 animals. Joint torques were measured with custom-built force sensors with strain gauges. Prior to the experiments, the measured force [mN] was calibrated from the force-sensor value [V] with weights of known mass (0.2–5 g). Two small insect pins attached to the tip of the force transducer held the middle part of the femur of the middle leg (*Figure 1A* right). The length between the ThC or CTr joints and the attachment point at the femur was measured and used as the moment arm for the calculation of torque.

### Electrical stimulation

We developed a custom-built electrical stimulator for stimulating muscles (*Figure 1A* left). An extension circuit board was designed for Raspberry Pi 3 B+ (Raspberry Pi Foundation), including isolated 8-channel PWM signal outputs. The parameters of the PWM signals, for example, frequency (1–120 Hz) and duty ratio (0 to 100%), were changed using a Raspberry Pi microprocessor. The amplitude of the output voltage (0–9 V) was changed using variable resistors on the circuit board, which enabled the investigation of the effects of these parameters on torque generation due to muscle stimulation. In this study, we systematically analyzed the joint torques generated by muscle contraction as induced by bursts of PWM pulses. To do so, we varied the amplitude [V], frequency [Hz] and duty ratio [%] of the PWM-signal, and identified the combinations that most effectively and repeatedly produced torque.

For one trial of the stimulation experiments, the frequency, duty ratio, and amplitude (voltage) of the PWM signals were not changed, but the burst duration $T_i$ of the signals was changed (*Figure 1C* top). Owing to the slow activation dynamics of an insect muscle, burst duration is one of two key parameters for controlling isometric muscle-contraction force because the muscle essentially acts as a second-order low-pass filter (*Harischandra et al., 2019*). The pulse frequency is the other key parameter, which can be held constant because burst duration alone is sufficient to effectively control joint torques in the range of 0–1.0 [s].

## Electrode implementation

A pair of stimulation electrodes was implanted into each muscle through two small holes in the cuticle. Holes were pierced using an insect pin, and wires were fixed with dental glue (*Figure 1B*). The stimulation electrodes were thin silver wires (A-M Systems, diameter = 127 μm, without insulation; 178 μm with Teflon insulation). The insulation at the end of the silver wire was removed, and the wires were implanted. The other end of the stimulation electrode was connected to the output of the electrical stimulator. The correctness of the electrode implantation was verified through triggered *resistance reflexes*, which are responses to imposed movements of the ThC and CTr joints for the corresponding muscles.

## Data collection

We determined the parameter set with a frequency of 50 Hz and a duty ratio of 30%, which would allow continuous and effective torque generation in a pre-experiment. We performed electrical stimulation experiments in the following order: (i) first, we selected one of the three muscles (protractor, retractor, levator) to be stimulated in each stick insect and (ii) performed electrical stimulation of the selected muscle more than 50 times at 1 V (50 Hz, 30%). The duration of the electrical stimulation, $T_i$, was set manually and randomly; this was followed by a (iii) 3-min-resting-period to reduce the effect of muscle fatigue (the resting period was determined in the pre-experiment). (iv) We then performed electrical stimulations at 2 V, 3 V, and 4 V for more than 50 times each; note that each stimulation was preceded by a 3-min-resting-period. (v) A voltage from 1 to 4 V that effectively generated the torque for the corresponding muscle was selected. (vi) Following this selection, we conducted electrical stimulation experiments for each combination of frequency (30 Hz, 50 Hz, 70 Hz, 90 Hz, and 110 Hz) and duty ratio (10%, 30%, 50%) for more than 50 times, with a resting time of 3 min between each condition. (vii) The next muscles were selected depending on the condition of the stick insect and within the time constraints, and we repeated steps (ii)–(vi) and recorded the data. (viii) The individuals were changed (on another day), and steps (i)–(vii) were repeated. We collected 10 individuals ($N = 10$ in *Table 1*) for each muscle using this procedure. Notably, even after such a large number of electrical stimulations of the muscles, we did not observe any significant biological damage to the stick insect nor any fatigue or warm-up effects.

## Data analysis

To investigate the dependence of externally induced joint torques by electrostimulating one out of the three leg muscles, we measured the force generated at the attachment point and multiplied it with the known moment arm as follows: (1) For different burst duration $T_i$ we estimated peak-to-peak sensor values $S_{p2p}(i)$ [V] (*Figure 1C* left). (2) Applying the conversion factor obtained from the previous calibration, we obtained peak-to-peak force change [mN] in response to stimulation. (3) The force was then multiplied with the measured moment arm [mm] to obtain the joint torque $\tau_i$ [μNm] (*Figure 1C* right).

## Widely applied information criterion (WAIC)

We compared the predictive performances of the formulated models using the mathematical index WAIC (*Watanabe, 2018Watanabe, 2005*; *Watanabe, 2010a*; *Watanabe, 2010b*). The WAIC is a measure of the degree to which an estimate of the predictive distribution is accurate relative to the true distribution (*Watanabe, 2018*). Essentially, it is based on the difference between the information conveyed by the mean and that conveyed by the variance. This difference is negative if the term corresponding to the mean exceeds that corresponding to the variance. The smaller (or more negative) the WAIC index, the higher the predictive value of the model variant.

When calculating the WAIC index for a hierarchical model, several calculation methods can be used, depending on the definition of the predictive distribution, that is, the type of unknown data distribution being predicted (*Watanabe, 2018*). We were interested in predicting muscle properties with electrostimulation for a new, additional animal, not including experimental date, to enable individualized leg 'control'. From this perspective, we constructed a new distribution of the predictive parameters of a new animal by marginalizing intermediate parameters assigned to each hierarchical model (models 1–2, 2–2, 2–3, 2–4; *Watanabe, 2018*; *Wakita et al., 2020*; *Harada et al., 2020*). This allows for a fair comparison of the prediction performance of hierarchical and non-hierarchical models. Referring to the method from the previous studies (*Wakita et al., 2020*; *Harada et al., 2020*), the WAIC was computed by numerical integration with MCMC (Markov Chain Monte Carlo) samples by using Simpson's law and the 'log_sum_exp' function provided by Stan (*Stan Development Team, 2023*).

From the models described above, the model with the smallest WAIC value was considered the most appropriate predictive model in terms of predictivity for a new animal.

## Acknowledgements

This work was supported by JSPS KAKENHI Grant-in-Aid for Scientific Research on Innovative Areas "Science of Soft Robot" project (JP21H00317), a Grant-in-Aid for the Promotion of Joint International Research (Fostering Joint International Research) (JP17KK0109), a Grant-in-Aid for Scientific Research (A) (JP23H00481), and the Tateishi Science and Technology Foundation (2023, Reserch Grant A, 2231006).

---

## Additional information

### Funding

| Funder | Grant reference number | Author |
|---|---|---|
| Japan Society for the Promotion of Science | JP21H00317 | Dai Owaki |
| Japan Society for the Promotion of Science | JP17KK0109 | Dai Owaki |
| Tateishi Science and Technology Foundation | Research Grant A,2231006 | Dai Owaki |
| Japan Society for the Promotion of Science | JP23H00481 | Dai Owaki |

The funders had no role in study design, data collection and interpretation, or the decision to submit the work for publication.

### Author contributions

Dai Owaki, Conceptualization, Resources, Data curation, Software, Formal analysis, Funding acquisition, Validation, Investigation, Visualization, Methodology, Writing - original draft, Project administration, Writing - review and editing; Volker Dürr, Conceptualization, Resources, Supervision, Validation, Investigation, Writing - review and editing; Josef Schmitz, Resources, Data curation, Supervision, Validation, Investigation, Methodology, Project administration, Writing - review and editing

### Author ORCIDs

Dai Owaki (ID) http://orcid.org/0000-0003-1217-3892

### Ethics

At present, animal care regulations do not need to be considered for insect research at Bielefeld University and Tohoku University. We strongly agree with the responsibility and ethical issues discussed by Xu et al., 2020c regarding animal experiments, both for invertebrates that do not require specific approval or for vertebrates that do. We conducted experiments on stick insects (Carausius morosus) following the principles of harm minimization, precaution, and the 4Rs (reduction, replacement,

refinement, and reproduction) at the individual level: (1) Reduction: We set N=10 as the maximum number of insects in each muscle in the experiment, which was considered the minimum number necessary to obtain statistically significant results. (2) Replacement: Although we previously surveyed various findings on the force characteristics generated in the muscles of stick insects based on nerve and muscle electrical stimulation (Blüme et al., 2012c,a,b; Harischandra et al., 2019), we still needed to conduct electrical stimulation experiments on animals. (3) Refinement: Preliminary experiments were conducted on a few stick insects to determine parameters that would not affect their behaviors or lives. Because stick insects do not actively walk in daily life, we did not use anesthesia to insert the electrodes. Measurements were performed in a manner that minimized potential pain, suffering, and distress. Even after a large number of electrical stimulations, we found no effect on insect behaviors; they resumed their normal activities once they returned to their breeding boxes in the colony. (4) Reproductivity: In subsequent studies, we conducted two experiments in which only the insect body was mounted: (i) an electric stimulation experiment for one leg in which all legs were in the air (no contact with the ground) and (ii) an electric stimulation experiment for one leg in which all legs were on the ground and generated gait patterns. The experiments yielded similar results under different conditions, reporting further findings using similar experimental protocols (paper in preparation). Furthermore, the authors completely agree that future research on cyborg insects, which may push the boundaries that are yet to be entirely considered by ethicists and legislators, will require careful ethical considerations of both animal welfare and social consequences.

## Decision letter and Author response
Decision letter https://doi.org/10.7554/eLife.85275.sa1
Author response https://doi.org/10.7554/eLife.85275.sa2

## Additional files

### Supplementary files
• MDAR checklist

### Data availability
We have now made our data and code (Figures 2–5, Appendix 1—Figures 1 and 2) accessible to the following link, ensuring that they are available for scrutiny and that our approach can be replicated by others. https://doi.org/10.5061/dryad.wpzgmsbsw.

The following dataset was generated:

| Author(s) | Year | Dataset title | Dataset URL | Database and Identifier |
|---|---|---|---|---|
| Owaki D | 2023 | Data from: A hierarchical model for external electrical control of an insect, accounting for inter-individual variation of muscle force properties | https://doi.org/10.5061/dryad.wpzgmsbsw | Dryad Digital Repository, 10.5061/dryad.wpzgmsbsw |

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

## Appendix 1

## Effect of frequency and duty ratio of PWM

Appendix 1 figure (*Appendix 1—figure 1*) 1 presents the variation of parameters and of the muscle characteristics with the PWM frequency for the six Bayesian models. The results indicate the following: (1) increases with frequency, but there exists an optimal frequency for each muscle; (2) is independent of the frequency; and (3) is affected by individual differences, whereas exhibits cross-individual consistency. In this study, we employed a frequency of 50 Hz, which had minimal effect on the individual differences in .

Furthermore, *Appendix 1—figure 2* presents the variation in the parameters and for the six Bayesian models as a function of the PWM duty ratio. The results reveal the following: (1) linearly increases with the duty ratio; (2) is independent of the duty ratio; and (3) is affected by individual differences, whereas is consistent across individuals. We employed an intermediate duty ratio of 30%, which yielded consistent data.

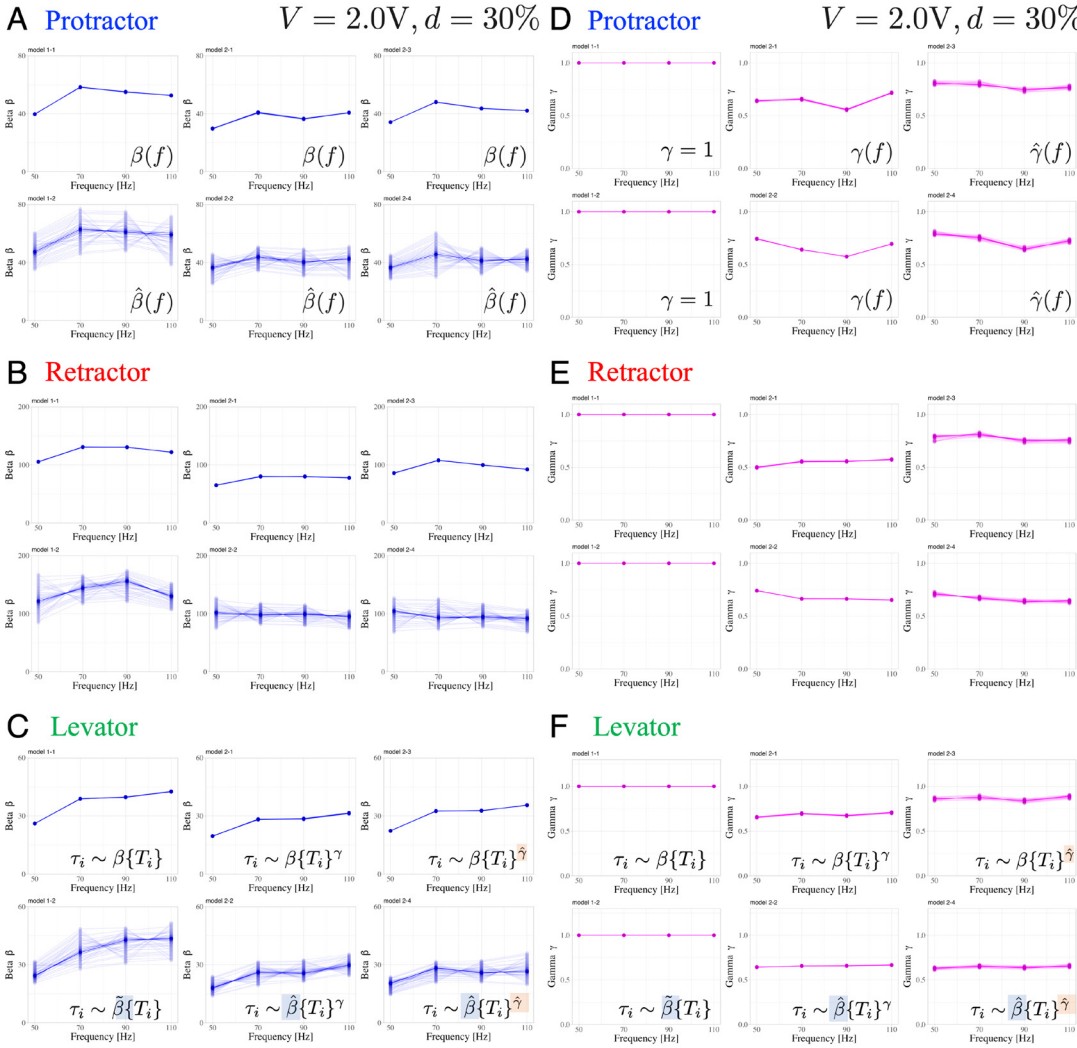

**Appendix 1—figure 1.** Dependence of muscle property parameters on the applied frequency of the PWM signals (2.0 V and 30% duty ratio) for the six models. The left graphs (**A**), (**B**), and (**C**) present the value of $\beta$ estimated for an applied frequency ranging from 50 to 110 Hz for the six models. (**A**) and (**D**), (**B**) and (**E**), and (**C**) and (**F**) illustrate the protractor, retractor, and levator stimulations, respectively. In (**A**) to (**C**), the upper and lower panels refer to the non-hierarchical (1-1, 2-1, 2-3) and hierarchical (1-2, 2-2, 2-4) models for,$\beta$ respectively. The right graphs (**D**), (**E**), and (**F**) present the value of $\gamma$ estimated for the applied voltage changes. In (**D**) to (**F**), the left panel refers to the *Appendix 1—figure 1 continued on next page*

*Appendix 1—figure 1 continued*

linear models ($\gamma = 1$, 1–1, 1–2), and the middle and right panels refer to the non-hierarchical models (2-1, 2-2) and hierarchical (2-3, 2-4) models for, $\gamma$ respectively. Source code and data are available on Dryad (Figure6-7.zip, https://doi.org/10.5061/dryad.wpzgmsbsw).

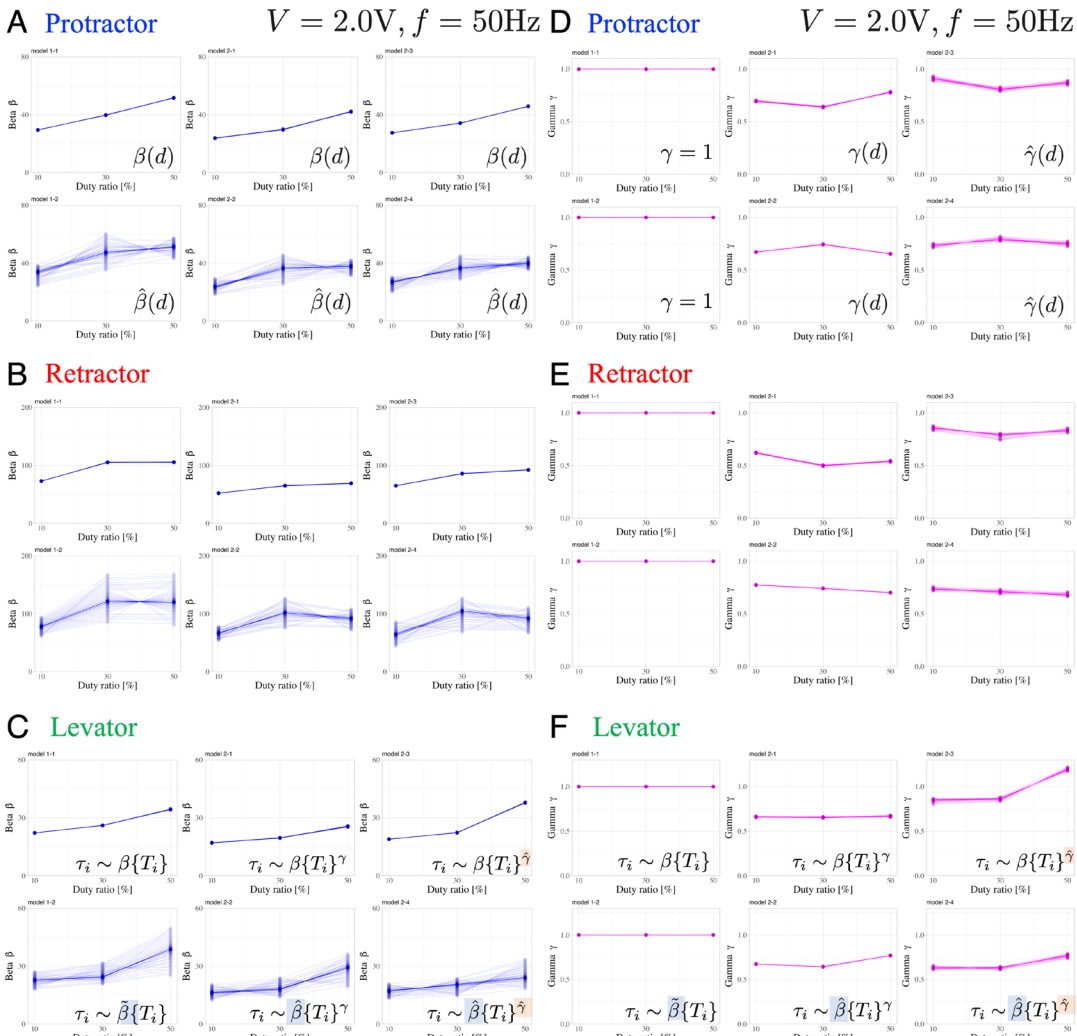

**Appendix 1—figure 2.** Dependence of muscle property parameters on the applied duty ratio of the PWM signals (2.0 V and 50 Hz) in the six models. The left graphs (**A**), (**B**), and (**C**) present the estimation for $\beta$ with the applied duty ratio varied from 10 to 30% for the six models. (**A**) and (**D**), (**B**) and (**E**), and (**C**) and (**F**) illustrate the protractor, retractor, and levator stimulations, respectively. In (**A**) to (**C**), the upper and lower panels present the non-hierarchical (1-1, 2-1, 2-3) and hierarchical (1-2, 2-2, 2-4) models for $\beta$, respectively. The right graphs (**D**), (**E**), and (**F**) present the estimation for $\gamma$ under the applied voltage changes. In (**D**) to (**F**), the left panel shows linear models ($\gamma = 1$, 1–1, 1–2), and the middle and right panels illustrate the non-hierarchical (2-1, 2-2) and hierarchical (2-3, 2-4) models for $\gamma$, respectively. Source code and data are available on Dryad (Figure6-7.zip, https://doi.org/10.5061/dryad.wpzgmsbsw).

## Relationship between generated joint torque and body morphology

We also examined the relationship between the joint torque generated by electrical muscle stimulation and body morphology, though in a separate sample of $N = 9$ individuals (*Appendix 1—figure 3*) that was different from the sample used for *Figures 2–5*, *Appendix 1—figures 1 and 2*. This is because experiments for *Figures 2–5*, *Appendix 1—figures 1 and 2* did not log size data, and experiments with suitable data on animal did not cover all three leg muscles (retractor, retractor, and levator). We considered the femur segment length (i.e., the length between the ThC

and FTi joints) of the stimulated middle leg and the body length (i.e., the length from head to tail) as body morphology features. Both lengths were measured from top-view videos of stick insects. As the characteristic parameters of joint torque, we considered the maximum joint torque $T_{max}$ [μNm] during electrical stimulation of the protractor and retractor muscles, and the average value of $\beta$ [μNm/s] in the linear models (model 1–2) for each individual. Note that there were no differences in joint torque characteristics between the figures (*Figures 2–5*, *Appendix 1—figures 1 and 2*) and *Appendix 1—figure 3*.

*Appendix 1—figure 3A* shows the correlation coefficients (color: purple = 1 to orange = –1) and p-values (numbers in the panel) between femur length, body length, $T_{max}$, and average $\beta$ for the protractor and retractor muscles. Statistically significant correlations ($p < 0.05$) were found only between either length measured (*Appendix 1—figure 3B*, $r = 0.669, p = 0.0487$) and $T_{max}$ and $\beta$ for the protractor ($r = 0.869, p = 0.00233$) and retractor ($r = 0.866, p = 0.00251$), respectively. Between femur length and joint torque features, we found a negative correlation for protractor $T_{max}$ and $\beta$ (upper left of *Appendix 1—figure 3C*), and a positive correlation for retractor $T_{max}$ and $\beta$ (upper right of App.*Figure 3C*). We likewise observed weak positive correlations between the body length and torque features for both the protractor (bottom left of App.*Figure 3C*) and retractor (bottom right of App.*Appendix 1—figure 3C*). Thus, no consistent pattern of correlation was found between individual differences in generated joint torque and bodily characteristics in this experimental data ($N = 9$ in App.*Figure 3*). Nonetheless, because weak correlations were observed, bodily characteristics may be considered as the input of a more precise prediction model that accounts for individual differences.

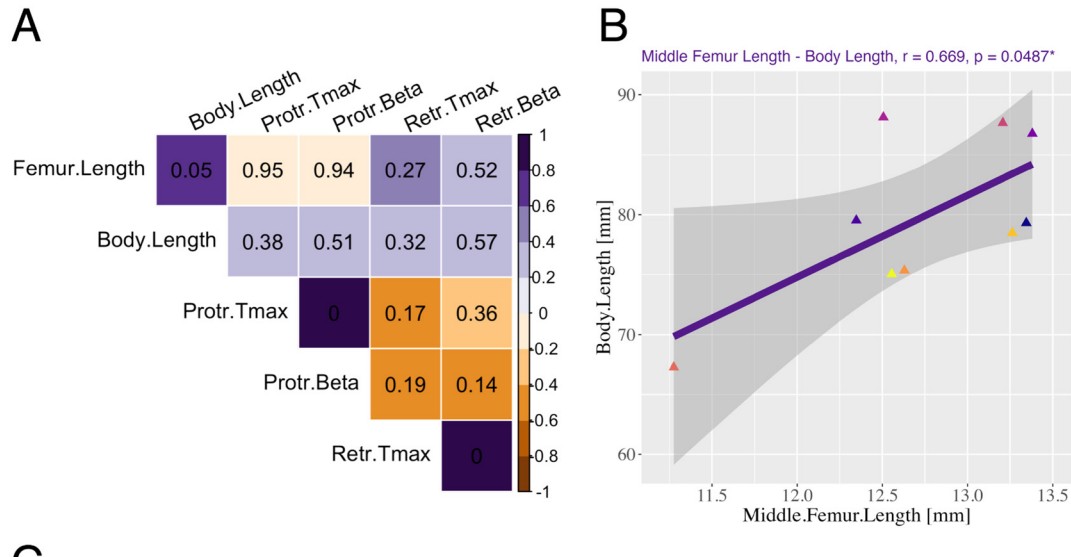

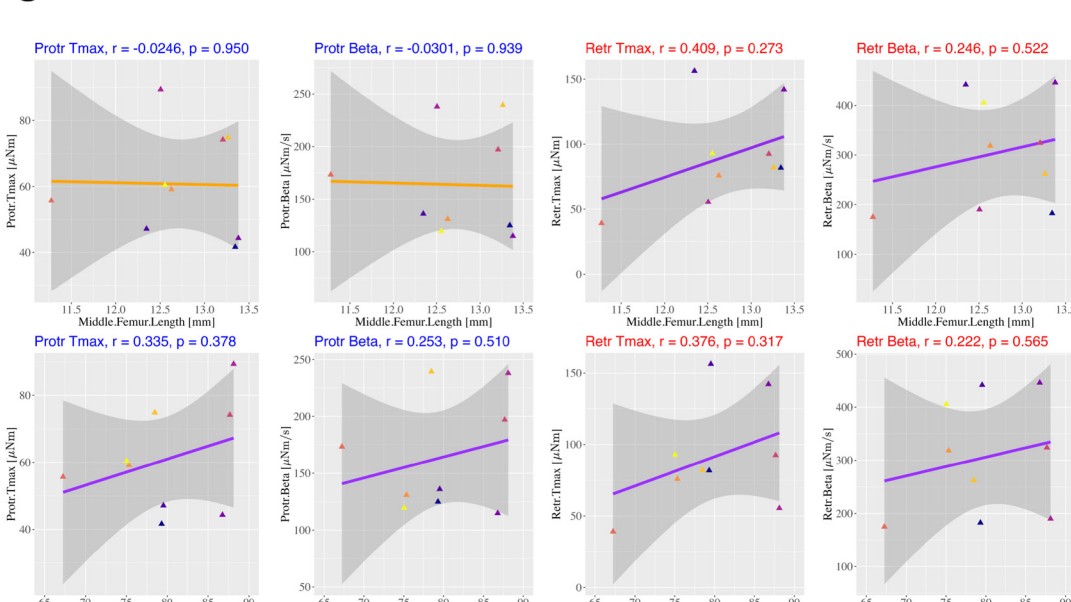

**Appendix 1—figure 3.** Relationship between generated joint torque and body morphology. (**A**) Correlation coefficients (color: purple = 1 to orange=-1) and $p$-values (numbers in the panel) between middle femur length, body length, Tmax, and the averaged $\beta$ for protractor and retractor muscles. (**B**) Linear regression between middle femur length and body length. The purple line represents the linear regression line. (**C**) Linear regression between middle femur length (upper)/body length (lower) and $T_{max}$ and $\beta$ for protractor (left) and retractor (right) muscles, respectively. For (**B**) and (**C**), color differences in plot points indicate individual differences ($N = 9$), and the gray area represents the 95% confidence interval. The correlation coefficient $r$ and $p$-value are indicated in each panel. These data are for future follow-up studies and cannot be disclosed.

