## [Editor Report]

This valuable work presents new results to characterize the relationship between electrical excitation and torque generation in stick insect joints. The evidence supporting this work is a series of torque-voltage measurements across individuals. The strength of evidence is compelling in supporting the outcomes.

---

## [Decision Letter]

**Decision letter after peer review:**

Thank you for submitting your article "A hierarchical model for external electrical control of an insect, accounting for inter-individual variation of muscle force properties" for consideration by *eLife*. Your article has been reviewed by 3 peer reviewers, and the evaluation has been overseen by a Reviewing Editor and Anna Akhmanova as the Senior Editor. The following individual involved in the review of your submission has agreed to reveal their identity: Nick Gravish (Reviewer #2).

Essential revisions:

1) The clarity of this work suffers from its structure: the models (and the parameters within) are central to the results of this study. The integration of data-driven modeling and experiment is the main reason this work is exciting! Yet, these are introduced far after the results are presented. While this is partially due to the section structure set forward, some basic aspects of the models and experimental system should be introduced prior to delineating the results in order to provide clarity.

2) The referees were concerned that it was not clear from the presentation of the results how substantial the contributions in this paper are to the field as a whole. The authors should better articulate the importance of their contributions, or, missing that, they should better explain what the challenges have been and what would need to be done to overcome them.

3) Along these lines, authors are missing an opportunity to make their work more impactful, by limiting the motivation and discussion to the domain of cyborgs, which is in itself important but quite a small field of research. There are many important animal locomotions, and even mechanical sensing, problems where this understanding is extremely relevant and useful. For example, stiffening the legs can help animals generate larger forces during locomotion in rough terrain, in behaviors that can benefit from higher forces (e.g., escape from predators, fighting between males during courtship), even in mechanical sensing (e.g., web-making spiders may modulate leg stiffness as part of its strategies to modulate how prey vibration is sensed by vibration sensors in its legs). A few studies that may help the authors appreciate and think about the broad implications:

a. Sponberg et al. (2011), A single muscle's multifunctional control potential of body dynamics for postural control and running, Phil. Trans. Royal Soc. B. 366(1570): 1592-1605.

b. Blickhan (1986), Stiffness of an arthropod leg joint, J. Biomechanics, 19(5), 375-384.

c. Wang et al. (2022), Cockroaches adjust body and appendages to traverse cluttered large obstacles, J. Exp. Biol., 225(10), jeb243605.

d. Mhatre et al. (2018), Posture controls mechanical tuning in the black widow spider mechanosensory system, bioRxiv, 484238.

Please comment on the relationship of the results in this study to the above line of research.

4) While it is interesting that inter-individual differences are important in the torque output from the joint, are these inter-individual differences related to any distinct differences among the insects studied (e.g., body mass, limb length, cross-sectional muscle area, and age all would likely influence torque)? While the referees are not advocating that all of the above parameters (age, size, etc) be added into a more complex model, they think it is important to provide any known information about the variance in individual size/age/etc, perhaps as a supplementary table.

5) Line 145 states that "Models 1-2 and 2-1 most accurately predicted the posterior predictive distribution.", is this a typo? The referees were under the impression that Models 1-2 and 2-2 are the best, as they are linear and nonlinear models with hierarchical slopes. In the paragraph starting at line 147 and the subsequent paragraph it is argued that while the nonlinear model 2-2 worked well, the linear model is still better. "The comparison of the linear model (model 1-2) with the nonlinear model (model 2-2) using the WAIC for all conditions (muscle type and applied voltage) resulted in lower values for the linear model." But certainly, both are quite close in WAIC, and the question is: might there be reasons from muscle physiology on stick insects to expect a non-linear model? While the linear model had the marginally lowest WAIC without any prior assumptions about the torque-duration curve, certainly much is known about the effect of stimulation on force production, and might including that information validate the non-linear model over linear? Alternatively, if the goal is to just model the data under 500ms stimulation because this is the relevant timescale for walking behavior (line 181), then the linear model is fine. But reading the manuscript the referees got the impression the goal was to best model the torque-voltage relationship, which would include the full excitation range and incorporates known information from muscle physiology. Please comment on these concerns and edit the manuscript as needed.

6) Figure 3 is a bit confusing, as this plot is meant to compare the experimental data with the hierarchical model distribution. However, all the model distributions across the 10 insects look identical. Wasn't the point of the hierarchical model that the slope parameter varies across individuals (isn't this what Figure 4 demonstrates?)? So, shouldn't the distributions and green fit lines all be different for the individuals? Please comment.

7) It is stated that 20 insects were tested, but all the plots show only 10. Is this just because the other 10 were not presented? Or were observations discarded from the other 10 insects for some reason? This is important to describe so that readers can assess the results.

8) What is the order of presentation of different voltages? It is stated that muscle fatigue should be negligible for under 50 stimulations, but the range of the 2V experiments alone is between 49-79 stimulations. So, were another ~50 stimulations performed at the three other voltages? And if so, was fatigue a possible issue?

9) Also, were there "warm-up" effects too where the muscle force increased with subsequent stimulations? It would be important to provide some characterization of this.

10) More information should be provided about the ordering of the different excitation experiments. The methods do not describe what the time duration between excitations was, how many were performed over what time period, etc. Additionally, it looks like four different voltage amplitudes were performed which I could only observe from figures 2 and 4. It would be beneficial to describe in detail the full sequence of data collection on an insect.

11) It is stated that muscle fatigue should be negligible for under 50 stimulations, but the range of the 2V experiments alone was between 49-79 stimulations. So, were another ~50 stimulations performed at the three other voltages? And if so, was fatigue a possible issue? Also, were there "warm up" effects too where the muscle force increased with subsequent stimulations? It would be useful to provide some characterization of this.

12) The authors also seem to be only addressing certain parameters rather than the potential adjustable parameters. PWM, voltage, and frequency are adjustable, but the paper only varied voltage and burst duration. It is unclear whether factors such as frequency (which has been shown to affect muscle force values) were investigated or not. If they were investigated in preliminary experiments, it would help if they were described; if not, it would also help to explain why, to help the readers understand why only burst duration and voltage were varied.

13) The data and code were not yet made available. The referees request access to both the data set and the code, as both are necessary to assess the reproducibility of this study.

14) Given the potential ethical considerations of 'cyborg control of insects,' the authors should discuss the potential ethical implications of extensions of their work with respect to animal welfare and other societal implications.

*Reviewer #1 (Recommendations for the authors):*

Overall, I think this is an interesting and useful study and that it will nicely move the field forward. The primary suggestion I have is a slight reorganisation as noted in the Public Review: while I understand that journal section structure puts some limitations on this (and while I agree that overly technical information should be placed so as not to disrupt the flow of the narrative), introducing some basic features of the experiments and models upfront (perhaps in a Table form) would be very helpful in understanding what the results mean. I also recommend moving Figure 5 before the results Figures.

The data and code were not yet made available (as far as I can tell), which is a bit disappointing. From the text, I understand they will be made available upon publication; but it is difficult to assess the reproducibility of this study without access to these as a referee.

*Reviewer #3 (Recommendations for the authors):*

1. The work itself seems not very substantial. It seems that the authors did relatively simple experiments, and just tried many different simple models to fit the data. It is not clear whether there is a substantial contribution. The authors should think harder about this and better articulate the contribution to the field with such a relatively simple study (as it appears). Or explain better what the challenges have been to better show why this initial first step is not as straightforward as it appears to be.

2. I think the author is missing an opportunity to make their work more impactful, by limiting the motivation and discussion to the domain of cyborgs, which is in itself important but quite a small field of research. There are many important animal locomotions and even mechanical sensing problems where this understanding is extremely relevant and useful. For example, stiffening the legs can help animals generate larger forces during locomotion in rough terrain, in behaviors that can benefit from higher forces (e.g., escape from predators, fighting between males during courtship), even in mechanical sensing (e.g., web-making spiders may modulate leg stiffness as part of its strategies to modulate how prey vibration is sensed by vibration sensors in its legs). A few studies that may help the authors appreciate and think about the broad implications:

a. Sponberg et al. (2011), A single muscle's multifunctional control potential of body dynamics for postural control and running, Phil. Trans. Royal Soc. B. 366(1570): 1592-1605.

b. Blickhan (1986), Stiffness of an arthropod leg joint, J. Biomechanics, 19(5), 375-384.

c. Wang et al. (2022), Cockroaches adjust body and appendages to traverse cluttered large obstacles, J. Exp. Biol., 225(10), jeb243605.

d. Mhatre et al. (2018), Posture controls mechanical tuning in the black widow spider mechanosensory system, bioRxiv, 484238.

3. The authors seem to be only addressing certain parameters rather than the potential adjustable parameters. PWM, voltage, and frequency are adjustable, but the paper only varied voltage and burst duration. It is unclear whether factors such as frequency (which has been shown to affect muscle force values) were investigated or not. If they were investigated in preliminary experiments, it would help if they were described; if not, it would also help to explain why, to help the readers understand why only burst duration and voltage were varied.

4. It is difficult to understand the Results and Discussion without reading the Method and Materials first. I know that *eLife* has Methods later, but the meaning of certain acronyms was not at least briefly explained until later in the paper, making it hard to understand when one reads it.

5. What are the resulting modelling equations generated for each? Is it possible to output the resulting modeling equations created from the Makrov Chain Monte Carlo method? It is difficult to see how they compare and are different from the simple linear and power equations that are used for 1-1 and 2-1.

a. What is the power function constant used for 2-1? It seems to be that \γ is 1, but doesn't that make it a linear function?

6. It is unclear how the author settled at the default parameters of the PWM signals to 2 V, 50 Hz, and 30% duty ratio.

7. For Figure 3, why is the prediction only compared with models 1-2? From what I gather, models 1-2 and 2-2 were the most accurate in predicting the posterior predictive distribution, why is specifically 1-2 addressed?

8. The intro addresses how inter-species variability can cause issues with the precise control of different animals. Is this issue addressed in this paper? It is not clear to me how this modelling can account for individual species variability considering the models only include variables for the burst duration and joint torque. Is the assumption that generating an appropriate model can lead to creating a robust feedback control system to control for interspecies variability?

9. The pictures of the experimental setup are confusing, it would be helpful if there was a schematic of the setup and some labels were given on where the muscles that were tested are located.

10. Not sure what the difference between hierarchical models and non-hierarchical models is, and where it is addressed.

11. Overall there are too many plots to understand, reducing the number of plots and increasing the font size on the plots will help increase the clarity and understanding of each figure.

Specific Comments:

1. Can you explain why there is a different number of simulations (n) for each animal? (Referring to Figure 3)

2. Unknown o? on line 349, not sure if hierarchical model o is a thing.

3. The labels for each of the y values and x values are very hard to see and are very blurry, it is hard to get a good sense of what these numbers mean for Figure 4, or what the y-axis and x-axis mean. Increasing the number font would be helpful for reading any of these graphs.

---

## [Author Response]

Essential revisions:1) The clarity of this work suffers from its structure: the models (and the parameters within) are central to the results of this study. The integration of data-driven modeling and experiment is the main reason this work is exciting! Yet, these are introduced far after the results are presented. While this is partially due to the section structure set forward, some basic aspects of the models and experimental system should be introduced prior to delineating the results in order to provide clarity.

Thank you for your valuable comment and suggestion. As correctly pointed out, Bayesian statistical modeling, the parameters of the models, and the experimental setup used to obtain the data are central to the results of this study. In light of this, we have restructured the manuscript by moving the "Experimental setup” (this is combined with “Burst duration and generated joint torque” and changed to "Joint torque measurements”), "Bayesian statistical modeling,” and "Model" subsections from the “Methods and Materials” section to the “Results” section. Additionally, we have adjusted the order of figures to align with this revised structure. We believe that these changes will significantly facilitate readers’ understanding of the study. The differences between the structure of the original version and that of the revised manuscript are summarized below:

[original paper]

Results

Burst duration and generated joint torque (Figure 1)

Comparison of model predictability (Figure 2)

Bayesian estimation of generated torque for a given burst duration (Figure 3)

Effect of an individual animal and applied voltage on muscle properties (Figure 4)

Methods and Materials

Animals

Experimental setup (Figure 5)

Electrical stimulation

Electrode implementation

Data analysis

Bayesian statistical modeling

Models

Widely Applied Information Criterion (WAIC)

[revised paper]

Results

Joint torque measurements (Figures 1 and 2)

Bayesian statistical modeling

Models

Comparison of model predictability (Figure 3)

Bayesian estimation of the generated torque for a given burst duration (Figure 4)

Effect of an individual animal and applied voltage on muscle properties (Figure 5)

Methods and Materials

Animals

Electrical stimulation

Electrode implementation

Data collection (new)

Data analysis

Widely applied information criterion (WAIC)

This comment significantly contributes to the reader's understanding of the paper. We greatly appreciate your kind suggestion.

2) The referees were concerned that it was not clear from the presentation of the results how substantial the contributions in this paper are to the field as a whole. The authors should better articulate the importance of their contributions, or, missing that, they should better explain what the challenges have been and what would need to be done to overcome them.

Thank you for your valuable feedback. As discussed in the second paragraph of the “Introduction” section in the first version of the manuscript, the inter-individual variability of animals is a central challenge that must be overcome, not only for cyborg control but also to understand animal locomotion. To emphasize the significance of our contribution more clearly, we have included the following sentences in the last paragraph of the “Introduction” section.

Introduction, last paragraph, L96-L104

“With regard to our general understanding of insect motor control, our study demonstrates that the dependency of joint torque on electrical stimulus duration is linear, despite nonlinear activation and contraction dynamics of insect muscle. Furthermore, the proposed hierarchical Bayesian model allows for a quick, simple and reliable measurement of the individual characteristics and, therefore, quantification of inter-individual differences. Whereas several studies have reported on inter-individual differences in neural (*Golowasch et al., 2002; Schulz et al., 2006;*) and muscle activity (*Horn et al., 2004; Brezina et al., 2005; Zhurov et al., 2005; Blümel et al., 2012c,a; Thuma et al., 2003; Hooper et al., 2006*), we propose how hierarchical Bayesian models may be used to harness inter-individual differences in insect locomotion research.”

3) Along these lines, authors are missing an opportunity to make their work more impactful, by limiting the motivation and discussion to the domain of cyborgs, which is in itself important but quite a small field of research. There are many important animal locomotions, and even mechanical sensing, problems where this understanding is extremely relevant and useful. For example, stiffening the legs can help animals generate larger forces during locomotion in rough terrain, in behaviors that can benefit from higher forces (e.g., escape from predators, fighting between males during courtship), even in mechanical sensing (e.g., web-making spiders may modulate leg stiffness as part of its strategies to modulate how prey vibration is sensed by vibration sensors in its legs). A few studies that may help the authors appreciate and think about the broad implications:a. Sponberg et al. (2011), A single muscle's multifunctional control potential of body dynamics for postural control and running, Phil. Trans. Royal Soc. B. 366(1570): 1592-1605.b. Blickhan (1986), Stiffness of an arthropod leg joint, J. Biomechanics, 19(5), 375-384.c. Wang et al. (2022), Cockroaches adjust body and appendages to traverse cluttered large obstacles, J. Exp. Biol., 225(10), jeb243605.d. Mhatre et al. (2018), Posture controls mechanical tuning in the black widow spider mechanosensory system, bioRxiv, 484238.Please comment on the relationship of the results in this study to the above line of research.

Thank you for your valuable comments. To highlight the importance and practical implications of our approach and findings, we have incorporated the following paragraph into the “Discussion” section, which introduces relevant studies related to animal locomotion, particularly in arthropods, and discusses their relevance to our study.

Discussion, sixth paragraph (new), L307-L332

“The contributions of this research are as follows: (1) this study demonstrates that compared to the nonlinear activation and contraction dynamics of insect muscles, the joint torque generated through electrical stimulation increases linearly with the duration of the stimulus, particularly during the stance and swing phases that characterize stick insect locomotion; (2) it introduces a hierarchical Bayesian model that allows for a reliable and simple description of the individual differences observed in neuromuscular system parameters. These contributions not only advance the field of insect cyborg control but also enhance our understanding of insect locomotion mechanisms. Animal locomotion is not solely governed by the brain and nervous system but also relies on the physical properties of the body and its interactions with the environment (*Chiel et al., 2009; Nishikawa et al., 2007*). Arthropods, in particular, effectively utilize mechanical properties and environmental interactions in their locomotion. Studies have revealed various related strategies, including the joint stiffness nonlinearity and hysteresis in spiders (*Blickhan, 1986*), generation of large motor outputs during escape maneuvers (*Card, 2012*) and posture stabilization (*Blickhan, 1986*) by adjusting the joint stiffness, mechanical sensing based on frequency characteristics that vary with the joint stiffness and posture in web-making spiders (*Blickhan and Barth, 2004; Mhatre et al., 2018*), transitions in movement patterns in response to mechanical interactions with the environment (*Othayoth et al., 2020*), and transitions in the coordinated movements of the body and legs (*Wang et al., 2022*) in cockroaches. Similarly, to elucidate the animal locomotion mechanisms emerging from such complex interactions, Sponberg et al. (*Sponberg et al., 2011b,a*) conducted experiments similar to those in our study by perturbing neural feedback through artificial interventions on muscle action potentials (MAPs) in cockroaches (*Blaberus discoidalis (L.)*). In follow-up studies, we further estimated the passive joint stiffness and analyzed the phase responses of stick insects during walking by accurately controlling the joint torque based on the linear stimulus duration-joint torque model derived in this study. We believe that these approaches will contribute to a deeper understanding of stick insect walking mechanisms, such as their use of two different stride lengths in response to their environment (*Theunissen and Dürr, 2013*).”

4) While it is interesting that inter-individual differences are important in the torque output from the joint, are these inter-individual differences related to any distinct differences among the insects studied (e.g., body mass, limb length, cross-sectional muscle area, and age all would likely influence torque)? While the referees are not advocating that all of the above parameters (age, size, etc) be added into a more complex model, they think it is important to provide any known information about the variance in individual size/age/etc, perhaps as a supplementary table.

We sincerely appreciate the valuable comments. The questions, where the observed inter-individual differences come from or what they are related to is an interesting question – though not at the heart of our study. To address your point, we examined the relationship between the joint torque generated by electrical muscle stimulation and body morphology as a supplementary topic. The data were collected from a different sample than that presented in the paper (*N = 9* in a follow-up study in 2019), simply because the video material for the 2018 sample used for Figures 2–5, Appendix 1 figures 1 and 2 were not suitable for measuring body size (unfortunately, we did not log body size in the lab book). In the 2019 sample we did not measure characteristics for all three muscles per animal, but the joint torque features of each muscle type were identical to those obtained in 2018. Owing to the fact that the data come from separate sets of experiments, we would like to emphasize this by making the additional figure (Appendix 1 figure 3), rather than a genuine part of the main paper (Figure 2-5, Appendix 1 figure 1 and 2). The results in Appendix 1 figure 3 show no consistent and little significant correlation between individual differences in generated joint torque and body size. However, because weak correlations were present, we now state that bodily characteristics may potentially be used as model input for further improvement of the prediction models. Further research is required to facilitate a more detailed analysis so as to reveal the factors contributing to the observed individual differences in torque generation.

In our revision, we have added the following section. We have also added Appendix 1 figure 3 to “Appendix 1” as supplementary material.

Appendix 1, Relationship between generated joint torque and body morphology, L695-L722

“We also examined the relationship between the joint torque generated by electrical muscle stimulation and body morphology, though in a separate sample of N = 9 individuals (App. Figure 3) that was different from the sample used for Figures 2–5, App.Figures1 and 2. This is because experiments for Figures2–5, App.Figures1 and 2 did not log size data, and experiments with suitable data on animal did not cover all three leg muscles (retractor, retractor, and levator). We considered the femur segment length (i.e., the length between the ThC and FTi joints) of the stimulated middle leg and the body length (i.e., the length from head to tail) as body morphology features. Both lengths were measured from top-view videos of stick insects. As the characteristic parameters of joint torque, we considered the maximum joint torque Tmax [Nmm] during electrical stimulation of the protractor and retractor muscles, and the average value of β [Nmm/s] in the linear models (model 1-2) for each individual. Note that there were no differences in joint torque characteristics between the figures (Figures 2–5, App.Figures1 and 2) and App. Figure 3.

App. Figure 3 A shows the correlation coefficients (color: purple = 1 to orange = -1) and *p*-values (numbers in the panel) between femur length, body length, Tmax, and average β for the protractor and retractor muscles. Statistically significant correlations (p < 0.05) were found only between either length measured (App. Figure 3 B, r = 0.669, p = 0.0487) and Tmax and β for the protractor (r = 0.869, p = 0.00233) and retractor (r = 0.866, p = 0.00251), respectively. Between femur length and joint torque features, we found a negative correlation for protractor Tmax and β (upper left of App. Figure 3 C), and a positive correlation for retractor Tmax and β (upper right of App. Figure 3 C). We likewise observed weak positive correlations between the body length and torque features for both the protractor (bottom left of App. Figure 3 C) and retractor (bottom right of App. Figure 3 C). Thus, no consistent pattern of correlation was found between individual differences in generated joint torque and bodily characteristics in this experimental data (N = 9 in App. Figure 3). Nonetheless, because weak correlations were observed, bodily characteristics may be considered as the input of a more precise prediction model that accounts for individual differences.”

5) Line 145 states that "Models 1-2 and 2-1 most accurately predicted the posterior predictive distribution.", is this a typo? The referees were under the impression that Models 1-2 and 2-2 are the best, as they are linear and nonlinear models with hierarchical slopes. In the paragraph starting at line 147 and the subsequent paragraph it is argued that while the nonlinear model 2-2 worked well, the linear model is still better. "The comparison of the linear model (model 1-2) with the nonlinear model (model 2-2) using the WAIC for all conditions (muscle type and applied voltage) resulted in lower values for the linear model." But certainly, both are quite close in WAIC, and the question is: might there be reasons from muscle physiology on stick insects to expect a non-linear model? While the linear model had the marginally lowest WAIC without any prior assumptions about the torque-duration curve, certainly much is known about the effect of stimulation on force production, and might including that information validate the non-linear model over linear? Alternatively, if the goal is to just model the data under 500ms stimulation because this is the relevant timescale for walking behavior (line 181), then the linear model is fine. But reading the manuscript the referees got the impression the goal was to best model the torque-voltage relationship, which would include the full excitation range and incorporates known information from muscle physiology. Please comment on these concerns and edit the manuscript as needed.

First, as the reviewer correctly pointed out, the statements in L145 and L151 (in the original manuscript) should refer to model 2-2, rather than model 2-1. We have addressed this error in the revised version of the manuscript.

Discussion, first paragraph, L230

“Models 1-2 and 2-2 provide the most accurate predictions of the posterior predictive distribution.”

Discussion, second paragraph, L236

“models 1-2 and 2-2, wherein only β was a hierarchical parameter, performed the best.”

Second, we would like to thank you for the critical and valid remarks. As correctly pointed out, the accuracy and selection of a model depend on the phenomenon to be explained or the target value/parameter to be estimated. Therefore, to determine the most appropriate model, we should clarify the target of our explanation. In the present study, one goal was to understand leg movement control in stick insects during walking. Considering that the time scale corresponding to the walking behaviors (stance and swing phases: ~500 ms) was the most relevant to our analysis, we concluded that a simple hierarchical linear model (model 1-2) was appropriate owing to its simplicity and low error, as discussed in the third paragraph of the “Discussion” section in the original paper. By contrast, the hierarchical nonlinear model (model 2-2) is more appropriate for longer time scales, such as the complete excitation range of muscles (~500 ms). Therefore, we have added the following sentences to the “Discussion” section:

Discussion, third paragraph, L276-L282

“[…] This suggests that the estimated stimulus-torque characteristic captures the natural dynamic properties of leg muscles during walking in terms of both the duration of excitation and maximum torque. However, the hierarchical nonlinear model (model 2-2) would be more appropriate for estimating properties related to longer time scales, such as those associated with the complete range of muscle excitation. Nevertheless, we emphasize once again that a key contribution of this study lies in demonstrating, based on experimental data, that the muscle property γ across the complete excitation range exhibits inter-individual variations and is independent of linear or nonlinear properties; hence, the weight β assigned to these properties represents individual differences.”

6) Figure 3 is a bit confusing, as this plot is meant to compare the experimental data with the hierarchical model distribution. However, all the model distributions across the 10 insects look identical. Wasn't the point of the hierarchical model that the slope parameter varies across individuals (isn't this what Figure 4 demonstrates?)? So, shouldn't the distributions and green fit lines all be different for the individuals? Please comment.

Thank you for bringing this to our attention. We apologize for the confusion. In the previous version of the paper, Figures 2 and 3 (revised Figures 3 and 4) displayed the same distribution for all individuals because they represented the predicted distribution of unknown animal characteristics based on the experimental data. However, we acknowledge your suggestion for depicting the predicted distribution for each individual based on the data. Therefore, in the revised version, we have modified Figure 4 to present the predicted distribution for each individual and revised the explanation in the "Bayesian estimation of generated torque for a given burst duration" section as follows:

Results, Bayesian estimation of the generated torque for a given burst duration, L206-L208

“[…] Figure 4 depicts the distributions predicted by the linear hierarchical model (model 1-2) for each individual by overlapping the experimental data shown in Figure 2.”

7) It is stated that 20 insects were tested, but all the plots show only 10. Is this just because the other 10 were not presented? Or were observations discarded from the other 10 insects for some reason? This is important to describe so that readers can assess the results.

Thank you for bringing this to our attention; we have clarified this point by revising the relevant content in the "Animals" subsection, as well as in Table 1, as follows:

Methods and Materials, Animals, L367-L371

“[…] The animals were raised under a 12h:12h light:dark cycle at a temperature of 23.9 ± 1.3 °C (mean ± S.D). All experiments were conducted at room temperature (20–24 °C). Table 1 lists the stick insects used in the electrical stimulation experiments. Owing to a combination of experimental failures and time constraints, we could not obtain stimulation data for all three muscles from the same animal on a single day. Therefore, we collected data from 10 animals (*N=10*) for each muscle through experiments with 20 animals.”

8) What is the order of presentation of different voltages? It is stated that muscle fatigue should be negligible for under 50 stimulations, but the range of the 2V experiments alone is between 49-79 stimulations. So, were another ~50 stimulations performed at the three other voltages? And if so, was fatigue a possible issue?

Thank you for your thoughtful comments. In the voltage-change experiments, we followed a specific order of voltage application, starting from 1 V and gradually increasing to 4 V, for each individual. This sequence was determined based on a prior experiment, where we applied voltages as high as ~9 V to assess the endurance voltage, confirming that applying voltages of 5 V or lower would be sufficient to tolerate a large number of stimulus experiments, even when conducted for several hours a day. Therefore, we ensured that we could continue experiments at 4 V or lower without causing significant damage to the stick insect muscles, even when several stimuli were applied. Additionally, we verified the absence of fatigue induced by the magnitude of the applied voltage in the 1–4 V range. Table 1 summarizes the number of electrical stimuli administered to each muscle in each individual.

Accordingly, we have included the above explanation in the “Results” section and added relevant data to Table 1.

Results, Effect of an individual animal and applied voltage on muscle properties, L211-L214

“Figure 5 presents the variations in the muscle characteristic parameters β and γ in response to changes in the applied voltage. In the voltage-change experiments, we followed a specific order of voltage application, gradually increasing from 1 V to 4 V, for each individual. Furthermore, we confirmed that applying voltages ranging between 1 and 4 V did not induce fatigue. Table 1 summarizes the number of electrical stimuli administered to each muscle in each individual. We determined the changes in β and γ with respect to the applied voltage by analyzing the experimental results using the six Bayesian models.”

9) Also, were there "warm-up" effects too where the muscle force increased with subsequent stimulations? It would be important to provide some characterization of this.

As presented in the revised Figure 2 (Figure 1 in the original paper), in this study, we did not observe any significant changes in the generation force, including “fatigue” or “warm-up,” during the electrical stimulation of the muscles. This observation was consistent across experiments where the voltage and frequency (see below) were varied under the same conditions, as well as in the before-and-after experiments. As mentioned in our response to comment 8), we defined the experimental conditions after confirming that no significant changes occurred in torque generation over time in our prior experiments. To clarify this matter, we have modified the following sentence in the revised manuscript:

Revised paper, Results, Joint torque measurements, L132-L135

“For a given set of PWM parameters, the generated torque characteristics remained almost constant during all stimulations under the same condition, suggesting that muscle fatigue or warm-up effects were negligible for at least n = 50 stimulations. Furthermore, we verified that no significant changes occurred in muscle characteristics owing to the pre- and post-experimental relationship.”

10) More information should be provided about the ordering of the different excitation experiments. The methods do not describe what the time duration between excitations was, how many were performed over what time period, etc. Additionally, it looks like four different voltage amplitudes were performed which I could only observe from figures 2 and 4. It would be beneficial to describe in detail the full sequence of data collection on an insect.

Thank you for your comments. To provide more detailed information on the experimental procedures for data collection, we have added a subsection "Data collection" to "Methods and Materials,” along with corresponding descriptions.

Methods and Materials, Data collection, L404-L422

"We determined the parameter set with a frequency of 50 Hz and a duty ratio of 30%, which would allow continuous and effective torque generation in a pre-experiment. We performed electrical stimulation experiments in the following order: (i) first, we selected one of the three muscles (protractor, retractor, levator) to be stimulated in each stick insect and (ii) performed electrical stimulation of the selected muscle more than 50 times at 1 V. The duration of the electrical stimulation, *T_i_*_,_ was set manually and randomly; this was followed by a (iii) 3-min-resting-period to reduce the effect of muscle fatigue (the resting period was determined in the pre-experiment). (iv) We then performed electrical stimulations at 2 V, 3 V, and 4V for more than 50 times each; note that each stimulation was preceded by a 3-min-resting-period. (v) A voltage from 1 to 4 V that effectively generated the torque for the corresponding muscle was selected. (vi) Following this selection, we conducted electrical stimulation experiments for each combination of frequency (30 Hz, 70 Hz, 90 Hz, and 110 Hz) and duty ratio (10%, 30%, 50%) for more than 50 times, with a resting time of 3 min between each condition. (vii) The next muscles were selected depending on the condition of the stick insect and within the time constraints, and we repeated steps (ii)–(vi) and recorded the data. (viii) The individuals were changed (on another day), and steps (i)–(vii) were repeated. We collected 10 individuals (*N=10* in Tab. 1) for each muscle using this procedure. Notably, even after such a large number of electrical stimulations of the muscles, we did not observe any significant biological damage to the stick insect nor any fatigue or warm-up effects."

11) It is stated that muscle fatigue should be negligible for under 50 stimulations, but the range of the 2V experiments alone was between 49-79 stimulations. So, were another ~50 stimulations performed at the three other voltages? And if so, was fatigue a possible issue? Also, were there "warm up" effects too where the muscle force increased with subsequent stimulations? It would be useful to provide some characterization of this.

These remarks are the same as those in comments 8) and 9), which are already addressed. Please refer to our previous responses.

12) The authors also seem to be only addressing certain parameters rather than the potential adjustable parameters. PWM, voltage, and frequency are adjustable, but the paper only varied voltage and burst duration. It is unclear whether factors such as frequency (which has been shown to affect muscle force values) were investigated or not. If they were investigated in preliminary experiments, it would help if they were described; if not, it would also help to explain why, to help the readers understand why only burst duration and voltage were varied.

Thank you for your constructive comments. We have thoroughly investigated the effects of the frequency and duty ratio in previous experiments, as described in the "Data collection" section. Based on your comment, we have included additional results in Appendix1 figures 1 and 2 of the revised manuscript. From a biological perspective, our findings suggest that the burst duration is a key variable, as discussed in the second paragraph of the “Discussion” section (L162–L166 in the original paper): "the generated torque depends much less on PWM voltage and frequency (*Blümel et al., 2012c; Harischandra et al., 2019*) than it depends on burst duration, suggesting the total number of subsequent input pules are important. This is indeed what would be expected for a slow insect muscle (*Blümel et al., 2012c*) that essentially "counts" incoming spikes within a given time window."

Changing the PWM frequency is comparable to varying the number of spikes within a given period, whereas changing the duty ratio is comparable to varying the average voltage during a given period. Therefore, from both the technical and cyborg control perspectives, the regulation of burst duration provides valuable insights into feasibility.

Consequently, we have incorporated the “Effect of frequency and duty ratio of PWM” in the “Appendix 1” section and in Appendix 1 figures 1 and 2, as well as the cyborg control technical perspective in the "Discussion" section of the revised paper.

Appendix 1, Effect of frequency and duty ratio of PWM, L653-L664

“Appendix 1 figure (App. Figure) 1 presents the variation of parameters β and γ of the muscle characteristics with the PWM frequency for the six Bayesian models. The results indicate the following: (1) β increases with frequency, but there exists an optimal frequency for each muscle; (2) γ is independent of the frequency; and (3) β is affected by individual differences, whereas γ exhibits cross-individual consistency. In this study, we employed a frequency of 50 Hz, which had minimal effect on the individual differences in β.

Furthermore, App. Figure 2 presents the variation in the parameters β and γ for the six Bayesian models as a function of the PWM duty ratio. The results reveal the following: (1) β linearly increases with the duty ratio; (2) γ is independent of the duty ratio; and (3) β is affected by individual differences, whereas γ is consistent across individuals. We employed an intermediate duty ratio of 30%, which yielded consistent data.”

Discussion, second paragraph, L254-L258

“[…]: The generated torque depends considerably less on the PWM voltage and frequency (*Blümel et al., 2012c; Harischandra et al., 2019*) than on the burst duration, suggesting that the total number of subsequent input pulses is important. This is indeed expected for a slow insect muscle (*Blümel et al., 2012c*) that essentially "counts" incoming spikes within a given time window. Compared to the nonlinear properties of muscle, we demonstrated that our monitoring of torques in an intact animal resulted in a linear characteristic (for intervals up to 500 ms) that would not be expected from isometric force measurements of isolated muscles. Furthermore, changing the PWM frequency was found to be comparable to changing the number of spikes over a given period, whereas changing the duty ratio was found to be comparable to varying the average voltage over a given period (see Appendix 1 figures 1 and 2). Therefore, from both technical and cyborg control viewpoints, the control of burst duration provides beneficial insights into feasibility.”

13) The data and code were not yet made available. The referees request access to both the data set and the code, as both are necessary to assess the reproducibility of this study.

We have now made our data and codes accessible to the following link, ensuring that they are available for scrutiny and that our approach can be replicated by others.

https://datadryad.org/stash/share/WXXJcMLl00KrfyEeZHi7rqfwoKmJd5GSVPQO7aKcQzk

14) Given the potential ethical considerations of 'cyborg control of insects,' the authors should discuss the potential ethical implications of extensions of their work with respect to animal welfare and other societal implications.

Thank you for your comments on this critical issue. We strongly agree with the responsibility and added a new ethics statement in which we discuss the following paper regarding animal experiments, whether on invertebrate animals that do not require specific approval or on vertebrate animals that do.

N. Xu et al., "Ethics of Biohybrid Robotic Jellyfish Modification and Invertebrate Research," preprint.org, doi: 10.20944/preprints202010.0008.v1, 2020.

Ethics statement, L462-L489

“At present, animal care regulations do not need to be considered for insect research at Bielefeld University and Tohoku University. We strongly agree with the responsibility and ethical issues discussed by (*Xu et al., 2020a*) regarding animal experiments, both for invertebrates that do not require specific approval or for vertebrates that do. We conducted experiments on stick insects (*Carausius morosus*) following the principles of harm minimization, precaution, and the 4Rs (reduction, replacement, refinement, and reproduction) at the individual level:

(1) Reduction: We set *N=10* as the maximum number of insects in each muscle in the experiment, which was considered the minimum number necessary to obtain statistically significant results.

(2) Replacement: Although we previously surveyed various findings on the force characteristics generated in the muscles of stick insects based on nerve and muscle electrical stimulation (*Blüme et al., 2012c,a,b; Harischandra et al., 2019*), we still needed to conduct electrical stimulation experiments on animals.

(3) Refinement: Preliminary experiments were conducted on a few stick insects to determine parameters that would not affect their behaviors or lives. Because stick insects do not actively walk in daily life, we did not use anesthesia to insert the electrodes. Measurements were performed in a manner that minimized potential pain, suffering, and distress. Even after a large number of electrical stimulations, we found no effect on insect behaviors; they resumed their normal activities once they returned to their breeding boxes in the colony.

(4) Reproductivity: In subsequent studies, we conducted two experiments in which only the insect body was mounted: (i) an electric stimulation experiment for one leg in which all legs were in the air (no contact with the ground) and (ii) an electric stimulation experiment for one leg in which all legs were on the ground and generated gait patterns. The experiments yielded similar results under different conditions, reporting further findings using similar experimental protocols (paper in preparation).

Furthermore, the authors completely agree that future research on cyborg insects, which may push the boundaries that are yet to be entirely considered by ethicists and legislators, will require careful ethical considerations of both animal welfare and social consequences.”